# Purine metabolism regulates DNA repair and therapy resistance in glioblastoma

Weihua Zhou[1,14], Yangyang Yao[1,2,14], Andrew J. Scott[1,3,14], Kari Wilder-Romans[1], Joseph J. Dresser[1], Christian K. Werner [1], Hanshi Sun[1], Drew Pratt[4], Peter Sajjakulnukit [5], Shuang G. Zhao[1], Mary Davis[1], Barbara S. Nelson[5], Christopher J. Halbrook[5], Li Zhang[5], Francesco Gatto [6], Yoshie Umemura[3,7], Angela K. Walker[8], Maureen Kachman [8], Jann N. Sarkaria [9], Jianping Xiong[2], Meredith A. Morgan[1,3], Alnawaz Rehemtualla[1,3], Maria G. Castro [3,10,11], Pedro Lowenstein [3,10,11], Sriram Chandrasekaran [3,12], Theodore S. Lawrence[1,3], Costas A. Lyssiotis [3,5,13] & Daniel R. Wahl [1,3✉]

Intratumoral genomic heterogeneity in glioblastoma (GBM) is a barrier to overcoming therapy resistance. Treatments that are effective independent of genotype are urgently needed. By correlating intracellular metabolite levels with radiation resistance across dozens of genomically-distinct models of GBM, we find that purine metabolites, especially guanylates, strongly correlate with radiation resistance. Inhibiting GTP synthesis radiosensitizes GBM cells and patient-derived neurospheres by impairing DNA repair. Likewise, administration of exogenous purine nucleosides protects sensitive GBM models from radiation by promoting DNA repair. Neither modulating pyrimidine metabolism nor purine salvage has similar effects. An FDA-approved inhibitor of GTP synthesis potentiates the effects of radiation in flank and orthotopic patient-derived xenograft models of GBM. High expression of the rate-limiting enzyme of de novo GTP synthesis is associated with shorter survival in GBM patients. These findings indicate that inhibiting purine synthesis may be a promising strategy to overcome therapy resistance in this genomically heterogeneous disease.

---

[1] Department of Radiation Oncology, University of Michigan, Ann Arbor, MI 48109, USA. [2] Department of Oncology, the First Affiliated Hospital of Nanchang University, Nanchang 330006 Jiangxi, PR China. [3] Rogel Cancer Center, University of Michigan, Ann Arbor, MI 48109, USA. [4] Department of Pathology, University of Michigan, Ann Arbor, MI 48109, USA. [5] Department of Molecular and Integrative Physiology, University of Michigan, Ann Arbor, MI 48109, USA. [6] Department of Biology and Biological Engineering, Chalmers University of Technology, 41296 Göteborg, Sweden. [7] Department of Neurology, University of Michigan, Ann Arbor, MI 48109, USA. [8] Biomedical Research Core Facilities, University of Michigan, Ann Arbor, MI 48109, USA. [9] Department of Radiation Oncology, Mayo Clinic, Rochester, MN 55902, USA. [10] Department of Neurosurgery, University of Michigan, Ann Arbor, MI 48109, USA. [11] Department of Cell and Developmental Biology, University of Michigan Medical School, Ann Arbor, MI 48109, USA. [12] Department of Biomedical Engineering, University of Michigan, Ann Arbor, MI 48109, USA. [13] Department of Internal Medicine, Division of Gastroenterology and Hepatology, University of Michigan, Ann Arbor, MI 48109, USA. [14] These authors contributed equally: Weihua Zhou, Yangyang Yao, Andrew J. Scott. ✉email: dwahl@med.umich.edu

Glioblastoma (GBM) is the most common aggressive adult primary brain tumor and is associated with profound genomic heterogeneity, which has limited therapy development. Work from The Cancer Genome Atlas (TCGA) and others have defined a diversity of molecular driver alterations in GBM[1,2]. Unfortunately, targeted therapies against these abnormalities have lacked efficacy in patients[3–6]. These disappointing results may be due to the profound intratumoral genomic heterogeneity of GBM. Indeed, single-cell and regional sequencing have shown that driver molecular events vary region-to-region and cell-to-cell within a single GBM[7–9]. This heterogeneity may explain why the only therapies that have improved survival in GBMs do not require a precise molecular alteration for activity: radiation (RT), temozolomide, surgery, and tumor treating fields[10].

RT is a critical treatment modality for GBM patients[11] and RT-resistance is a primary cause of recurrence and death. Fewer than 10% of patients with GBM live for 5 years and ~80% recur within the high dose RT field[12,13]. Thus, efforts to overcome RT-resistance are likely to improve outcomes in patients with GBM. Efforts to develop strategies to overcome RT-resistance have previously used large-scale genomic profiling data to define candidate oncogenic molecular alterations to target in combination with RT[14–16]. Due to the genomic heterogeneity of GBM and the disappointing performance of targeted therapies in GBM[3–6], we instead sought to define therapeutic strategies that could overcome RT-resistance independently of genotype.

Altered metabolism is a hallmark of cancers including GBM, is regulated by cell-intrinsic and -extrinsic factors, and could potentially regulate therapy resistance independently of genotype[17–21]. Importantly, disparate oncogenic alterations can activate common metabolic pathways, such as glycolysis[22]. Thus, GBMs with profound intratumoral genomic heterogeneity may have relatively common metabolic phenotypes that in turn mediate resistance to RT.

Here, we find that purine metabolites cause GBM RT-resistance by promoting the repair of RT-induced DNA double-stranded breaks (DSBs). Depleting purines with FDA-approved drugs radiosensitizes multiple GBM models in vitro and in vivo, including in an orthotopic patient-derived xenograft (PDX). High expression of inosine monophosphate dehydrogenase 1 (IMPDH1), a rate limiting enzyme in de novo GTP synthesis, is associated with inferior survival in GBM. These findings indicate that inhibiting purine synthesis may be a promising strategy to overcome therapy resistance in this genomically heterogeneous disease.

## Results

**Purines correlate with GBM RT-resistance**. To determine the characteristics of RT-resistance in GBM, we performed clonogenic survival assays on 23 immortalized GBM cell lines and found a wide distribution of intrinsic RT sensitivities (reported as Dmid, which is the mean inactivating dose of RT and defined as the area under the clonogenic cell survival curve[23]) (Fig. 1a). These lines were chosen because they had been genomically profiled by the Cancer Cell Line Encyclopedia (CCLE), were publicly available from cell line repositories (ATCC, DSMZ, and JCRB) and were amenable to both reproducible metabolomic analysis and the clonogenic survival assay in a uniform media (DMEM). None had mutations in isocitrate dehydrogenase 1 or 2 (IDH1/2) and thus were models of primary GBM. Intrinsic RT-resistance did not correlate with proliferation rate (Supplementary Fig. 1A) or cell cycle distribution (Supplementary Fig. 1B). RT causes DNA double-stranded breaks (DSBs) in DNA and the rapid phosphorylation of histone H2A variant H2AX, which

can be readily detected by immunoblot, flow cytometry or immunofluorescence[24]. Both RT-resistant (U87 MG and A172) and -sensitive cell lines (KS-1 and U118 MG) had high levels of γ-H2AX staining at 30 min and 2 h after RT (Fig. 1b) as measured by flow cytometry. However, γ-H2AX staining returned to baseline by 24 h after RT in the RT-resistant lines, while it remained persistently elevated in the RT-sensitive lines. Thus, RT-resistance in this GBM cell line panel is associated with an ability to effectively repair RT-induced DSBs.

Our group and others have postulated that abnormal metabolism in GBM may cause RT-resistance[25,26]. Using transcriptional data obtained from the CCLE, we asked whether expression of metabolic enzymes could predict for GBM RT-resistance. Consistent with our prior work[25], increased expression of IDH1 was associated with GBM RT-resistance (Supplementary Fig. 1C), presumably because this enzyme is an important source of NADPH in GBM. Glutamine synthetase (GLUL) expression was also associated with RT-resistance (Supplementary Fig. 1D), consistent with prior reports[27]. IDH3a, a critical mediator of oxidative ATP production through the TCA cycle, was instead associated with RT-sensitivity (Supplementary Fig. 1E). Gene set enrichment analysis revealed that three out of the top 10 most associated gene sets with RT-sensitivity were related to oxidative ATP production (Supplementary Fig. 1F). No such metabolic gene sets were found among the top 10 associated gene sets with RT-resistance (Supplementary Fig. 1G). This relative lack of actionable metabolic targets suggested a need to measure metabolism itself rather than the levels of metabolism-related transcripts.

We therefore performed targeted metabolomic analysis on each of the 23 GBM cell lines during unperturbed exponential growth (These data accompany this manuscript as Supplementary Data 1). Metabolites were grouped into corresponding pathways and correlations between pathway-level changes and RT-resistance were determined to identify metabolic phenotypes associated with RT-resistance. Downregulation of metabolites involved in de novo purine synthesis (inosinates and guanylates) were positively correlated with RT-sensitivity ($p < 0.03$, Fig. 1c and Supplementary Fig. 1H; Supplementary Data 1). Downregulation of the cytidine pathway was the third most-correlated metabolic pathway with RT-sensitivity, but was not statistically significant ($p = 0.08$). Thus, GBMs with lower nucleotide pools, especially purines, were more likely to be RT-sensitive.

We then asked how purine metabolism changed after cells were exposed to RT. Two hours after RT, a time point when DNA damage had occurred (Fig. 1b) but cells had not yet arrested or died (Supplementary Fig. 1B), both purine and pyrimidine metabolites increased in RT-resistant cell lines (Fig. 1d; Supplementary Data 1). RT-sensitive cell lines, however, increased neither purines nor pyrimidines following RT (Fig. 1d; Supplementary Data 1). In the post-RT setting, depleted guanylates was again the metabolic feature most correlated with RT-sensitivity ($p = 0.0001$, Fig. 1e; Supplementary Data 1). Decreased levels of metabolites related to glutathione, the primary cellular antioxidant, were significantly associated with RT-sensitivity ($p = 0.02$), which is consistent with the well-known oxidative mechanism by which RT kills cells[28]. Depletion of adenylates, the other main purine species, was also significantly associated with RT-sensitivity ($p = 0.0495$, Fig. 1e; Supplementary Data 1). Together, these results suggested that high levels of nucleobase-containing metabolites, especially purines, were related to GBM RT-resistance.

**Nucleoside supplementation protects GBMs from RT**. We next sought to determine whether the relationship between high

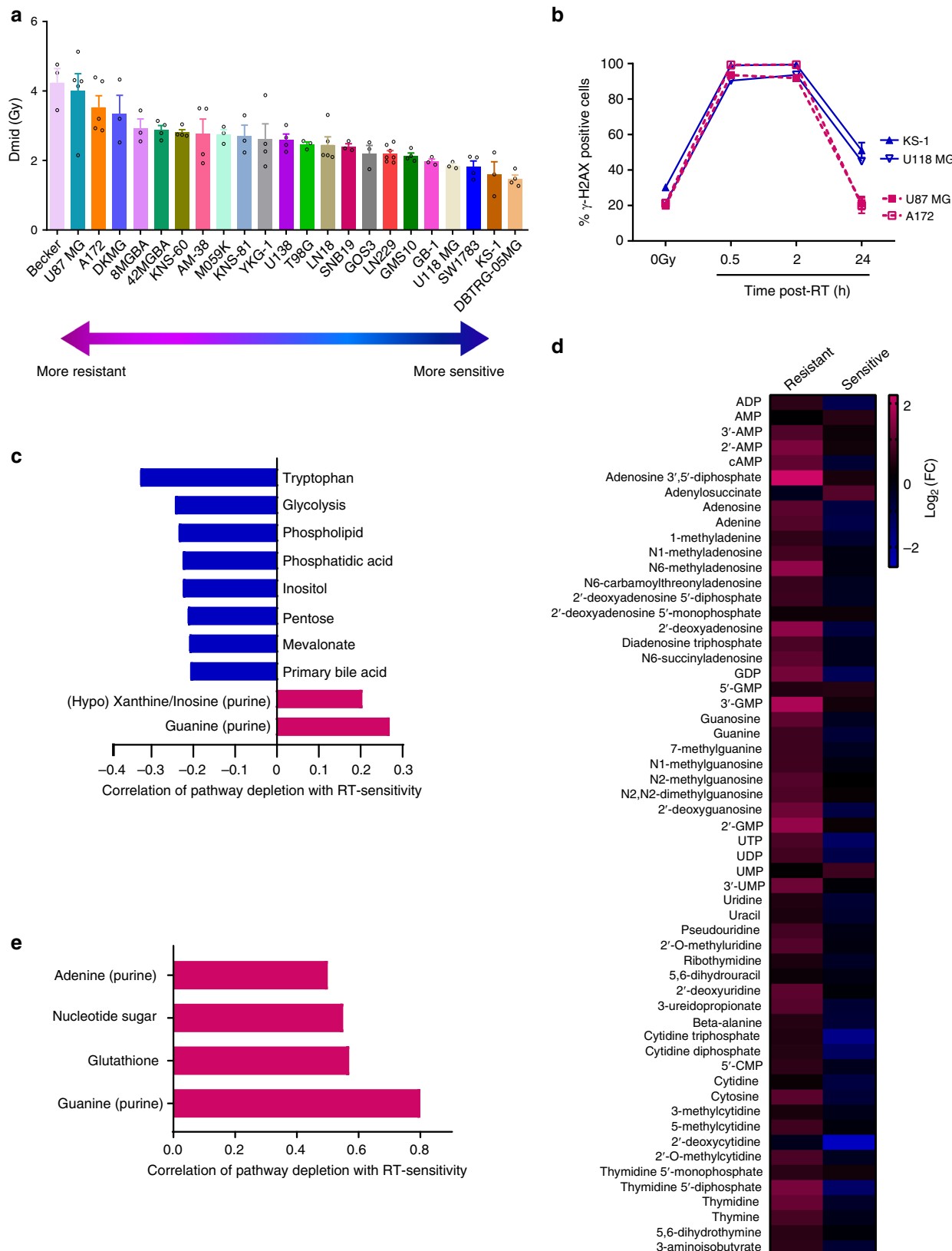

nucleobase-containing metabolites and RT-resistance in GBM was causal. Nucleotide-poor RT-sensitive GBM cell lines were supplemented with cell-permeable nucleosides (cytidine, guanosine, uridine, guanosine, and thymidine at concentrations 80–240 μM) and RT-sensitivity was determined by clonogenic assay (Fig. 2a). RT-sensitive GBM cell lines (U118 MG, DBTRG-05MG,

and GB-1; Fig. 1a) were protected from RT by exogenous nucleosides with Enhancement Ratios (ERs) ranging between 0.6 and 0.8 (Fig. 2b–d). ER of RT is defined as Dmid control divided by Dmid treatment. ERs below 1 indicate radioprotection and above 1 indicate radiosensitization. The RT-protection conferred by nucleosides was associated with decreased RT-induced DSBs.

**Fig. 1 Increased levels of nucleobase-containing metabolites are associated with RT-resistance in GBM. a** Clonogenic survival assays were performed on the indicated GBM cell lines to determine Dmid (the linear area under the clonogenic survival curve). Data are presented as mean ± SEM from 3 to 7 biologic replicates. **b** RT-resistant (U87 MG and A172) and RT-sensitive (KS-1 and U118 MG) GBM cell lines were irradiated with 8 Gy, followed by γ-H2AX flow cytometry analysis at 0, 0.5, 2, or 24 h following RT. Data are presented as mean ± SEM from 3 biologic replicates. **c** Metabolites were grouped into pathways and an average pathway-level correlation with RT-sensitivity was determined. Only the pathways significantly correlated with RT-sensitivity are shown. **d** Two RT-resistant (U87 MG and A172) and two RT-sensitive (KS-1 and U118 MG) GBM cell lines were irradiated with 8 Gy, and harvested 2 h after RT and analyzed by targeted LC-MS/MS (4 biologic replicates per cell line). Fold-change values for each metabolite were determined based on unirradiated matched cell line controls and then averaged for the two resistant (left) or sensitive (right) cell lines. **e** Pathways with downregulated metabolites post-RT that are significantly correlated with RT-sensitivity are shown (Pearson's correlation; $p = 0.0001$ for guanylates; $p = 0.02$ for glutathione; $p = 0.03$ for nucleotide sugar, $p = 0.0495$ for adenylates.). Source data are provided as a Source Data file.

Indeed, in all three sensitive cell lines, RT alone caused a peak of γ-H2AX foci within 30 min that did not return to baseline by 24 h, whereas treatment with exogenous nucleosides decreased γ-H2AX foci at 0.5, 2, 6, and 24 h following RT (Fig. 2e–g; Supplementary Fig. 2A–C).

Because DNA repair begins within seconds of damage[29], we were uncertain whether this decreased γ-H2AX staining meant that nucleosides were preventing the induction of DNA damage or facilitating its rapid repair. We therefore performed the alkaline comet assay[30], which measures physical DNA double-strand and single-strand breaks[31]. When performed on ice to arrest DNA repair, this assay measures only the induction of DNA damage. When performed at warmer temperatures and with longer incubation times after RT, this assay reflects both the induction and repair of DNA damage. Nucleosides did not change the amount of DNA damage induced when cells were irradiated on ice and harvested immediately (Fig. 2h, i; Supplementary Fig. 2D, E). However, exogenous nucleosides decreased the DNA damage that was present after repair was allowed to proceed at 37 °C for 0.5 and 4 h in two RT-sensitive GBM cell lines, DBTRG-05MG ($p < 0.01$ for 0.5 h and $p < 0.05$ for 4 h) and GB-1 ($p < 0.05$ for 0.5 h and 4 h, Fig. 2h, i; Supplementary Fig. 2D, E). Thus, supplementing nucleotide pools in RT-sensitive GBMs facilitates the repair of RT-induced DNA damage.

**Inhibiting GTP synthesis radiosensitizes RT-resistant GBMs.** Based on the above data, we next asked if lowering nucleotide pools would radiosensitize RT-resistant models of GBM. While nucleotides can either be salvaged or synthesized de novo, most GBMs are thought to rely on de novo nucleotide synthesis rather than nucleotide salvage[32]. Consistent with this hypothesis, a 4-h incubation with [15]N-amide glutamine labeled nearly half of the GMP, UMP, and CMP pools in the RT-resistant U87 GBM cell line, indicating substantial activity of de novo nucleotide synthesis. Less AMP was labeled (~30%), possibly due to the larger adenylate pool size (Supplementary Fig. 3A–D). These results suggested that inhibiting de novo nucleotide synthesis might have therapeutic utility in GBM.

Because guanylates were the metabolic pathway most associated with RT-resistance (Fig. 1c, e), we elected to intervene on this pathway (Fig. 3a). Mycophenolic acid (MPA) and its orally bioavailable prodrug mycophenolate mofetil (MMF) inhibit GTP synthesis by blocking the rate-limiting enzyme inosine monophosphate dehydrogenase (IMPDH). This intervention inhibits de novo GTP synthesis and also partially inhibits GTP salvage (if the salvaged base is hypoxanthine, Fig. 3a). MPA and MMF are FDA-approved to treat immune-mediated disorders, are being investigated as anticancer therapeutics and have favorable penetrance in the CNS where they are used for patients with conditions such as neurosarcoidosis[33–35]. Treatment with a clinically-relevant concentration of MPA (10 μM)[36] reduced GTP levels by more than tenfold, increased inosine monophosphate levels by more than tenfold and increased ATP levels 1.2-fold, consistent with inhibition of IMPDH and little GTP generation from guanine (Fig. 3a–d).

MPA radiosensitized two RT-resistant cell lines (treatment timeline shown as Supplementary Fig. 3E) in a concentration-dependent fashion (Fig. 3e, f). ERs ranged from 1.2 ± 0.1 in both U87 and A172 with 1 μM MPA treatment and 1.7 ± 0.3 in U87 MG and 2.3 ± 0.3 in A172 at 10 μM MPA, which is greater than the radiosensitizing effects of temozolomide, the standard radiosensitizer used in GBM[37]. The radiosensitizing effects of MPA were abrogated when cells were co-treated with exogenous nucleosides (ER: 1.0 ± 0.03 in U87 MG and 1.2 ± 0.1 in A172, Fig. 3e, f), indicating that MPA exerted its radiosensitizing effects through nucleotide depletion rather than off-target effects. Unlike in RT-sensitive GBM cell lines, exogenous nucleosides did not further protect RT-resistant GBM cell lines from RT (Fig. 3e, f). We speculate that RT-resistant cells are already rich in nucleotides (Fig. 1c, d), which limits the ability of further nucleoside supplementation to protect cells from RT.

The above findings were obtained in GBM cell lines that were resistant to RT in our initial profiling. While tractable for metabolomic and clonogenic survival assays, these immortalized GBM models may not fully recapitulate the histopathologic or molecular features of GBM tumors in patients[38]. We therefore confirmed our findings in primary patient-derived GBM neurosphere lines, referred to as HF2303 and MSP12[39]. These primary GBM cells form neurospheres when grown in serum-free conditions, are inherently resistant to RT and are thought to represent the cellular subtypes that mediate GBM recurrence after therapy[40,41]. Because neurospheres are not amenable to the clonogenic survival assay, we instead performed a long-term viability assay to assess the effects of RT. Primary neurospheres were treated as shown in Supplementary Fig. 3E. Treatment with MPA increased the RT-sensitivity of both HF2303 (ER: 1.4 ± 0.1) and MSP12 (ER: 1.7 ± 0.4 for MSP12) neurospheres. MPA-induced radiosensitization was reversed by exogenous nucleosides (ER: 0.8 ± 0.3 for HF2303 and 0.8 ± 0.1 for MSP12, Fig. 3g, h).

Because inhibition of IMPDH with MPA can inhibit both de novo and salvage GTP synthesis, it was unclear which pathway was responsible for mediating RT-resistance. We therefore asked if selectively inhibiting GTP salvage affected GBM RT-resistance. Blocking IMPDH-dependent GTP salvage (Fig. 3a) by depleting media hypoxanthine (from 30 μM in control media to absent in experimental media), did not sensitize U87 MG cells to RT (Supplementary Fig. 3F). We took an orthogonal approach in neurospheres because their culture conditions contain lower concentrations of hypoxanthine. Silencing *HPRT1*, the gene encoding HGPRT, which salvages both hypoxanthine (to form IMP) and guanine (to form GMP) (Fig. 3a), had no effect on the radiosensitivity of MSP12 neurospheres (Supplementary Fig. 3G, H). Hence, de novo GTP synthesis, rather than GTP salvage, appears to play a dominant role in mediating GBM RT-resistance.

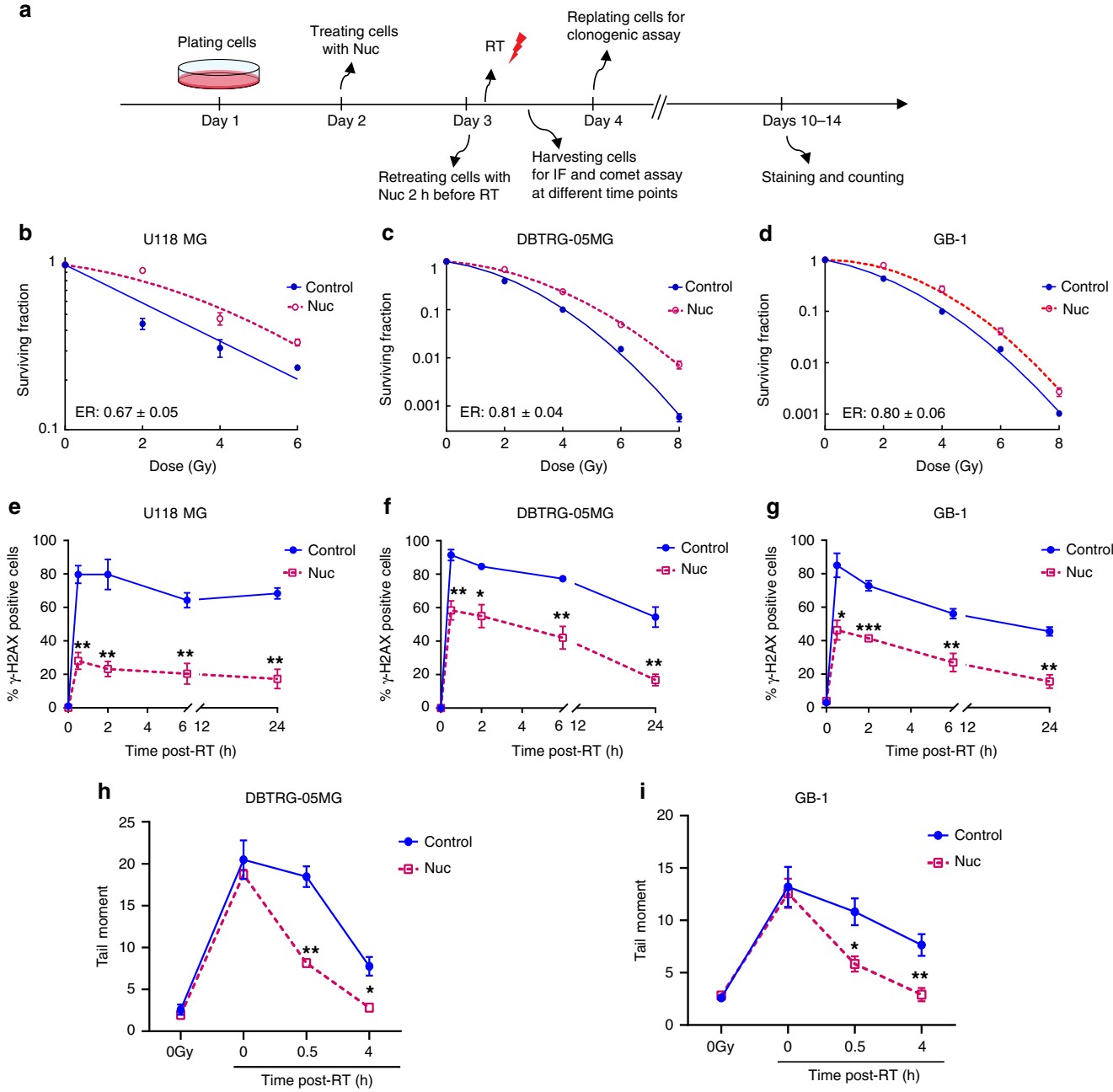

**Fig. 2 Supplementing nucleoside pools promotes DNA repair and RT-resistance in GBM. a** A schematic timeline of treatment in the RT-sensitive cell lines. U118 MG, DBTRG-05MG, or GB-1 cells were treated with exogenous pooled nucleosides (Nuc; 8×) for 24 h, and retreated with nucleosides 2 h before RT with indicated doses, followed by IF, comet assay, or clonogenic assay. **b**–**d** U118 MG, DBTRG-05MG, and GB-1 cells were treated as described in Fig. 2**a**, followed by clonogenic assay. ER indicates the Enhancement Ratio, which is the ratio of Dmid control and Dmid treatment. ER > 1 indicates radiosensitization while ER < 1 indicates radioprotection. Fig. **b**–**d** show representative figures from one of three biologic repeats for each cell line, each performed in technical triplicate. Error bars indicate mean ± SEM from technical triplicates from that single representative experiment. In the lower left of each graph, ER (mean ± SEM) from the three biologic replicates is shown. **e**–**g** Cells were treated as above and harvested for γ-H2AX foci IF staining at indicated times post-RT. Data are presented as mean ± SEM from 3 biologically independent experiments. p values of 0.5, 2, 6, and 24 h are 0.0021, 0.0050, 0.0044, and 0.0035 for Fig. **e**; 0.0071, 0.0134, 0.0069, and 0.0056 for Fig. **f**; 0.0140, 0.0007, 0.0093, and 0.0035 for Fig. **g**–**i**. DBTRG-05MG or GB-1 cells were treated as above and harvested at different time points for alkaline comet assay. Cells were irradiated and harvested on ice for the 0 h time point (4 Gy; 0 h). Data are presented as mean ± SEM from 3 (**h**) or 4 (**i**) biologically independent experiments. p values of 0, 0.5 and 4 h are 0.4996, 0.0019, and 0.0145 for Fig. **h**; 0.8050, 0.0152, and 0.0080 for Fig. **i**. Fig. **e**–**i**: *p < 0.05 and **p < 0.01, ***p < 0.001 compared with control. The p values indicated in Fig. **e**–**i** were obtained by two-tailed unpaired student's t test. Source data are provided as a Source Data file.

**Depleting GTP slows the repair of RT-induced DNA damage.**
We reasoned that GTP depletion may sensitize GBMs to RT by slowing DSB repair, much as nucleoside supplementation promoted DSB repair. Consistent with this hypothesis, the combination of MPA and RT increased γ-H2AX foci at various time points compared with RT alone in both U87 MG and A172 cells (p < 0.01, Fig. 4a; p < 0.05, Fig. 4b, Supplementary Fig. 4A, B). This increase (column 5 vs. 7, Fig. 4c, d) was rescued by the

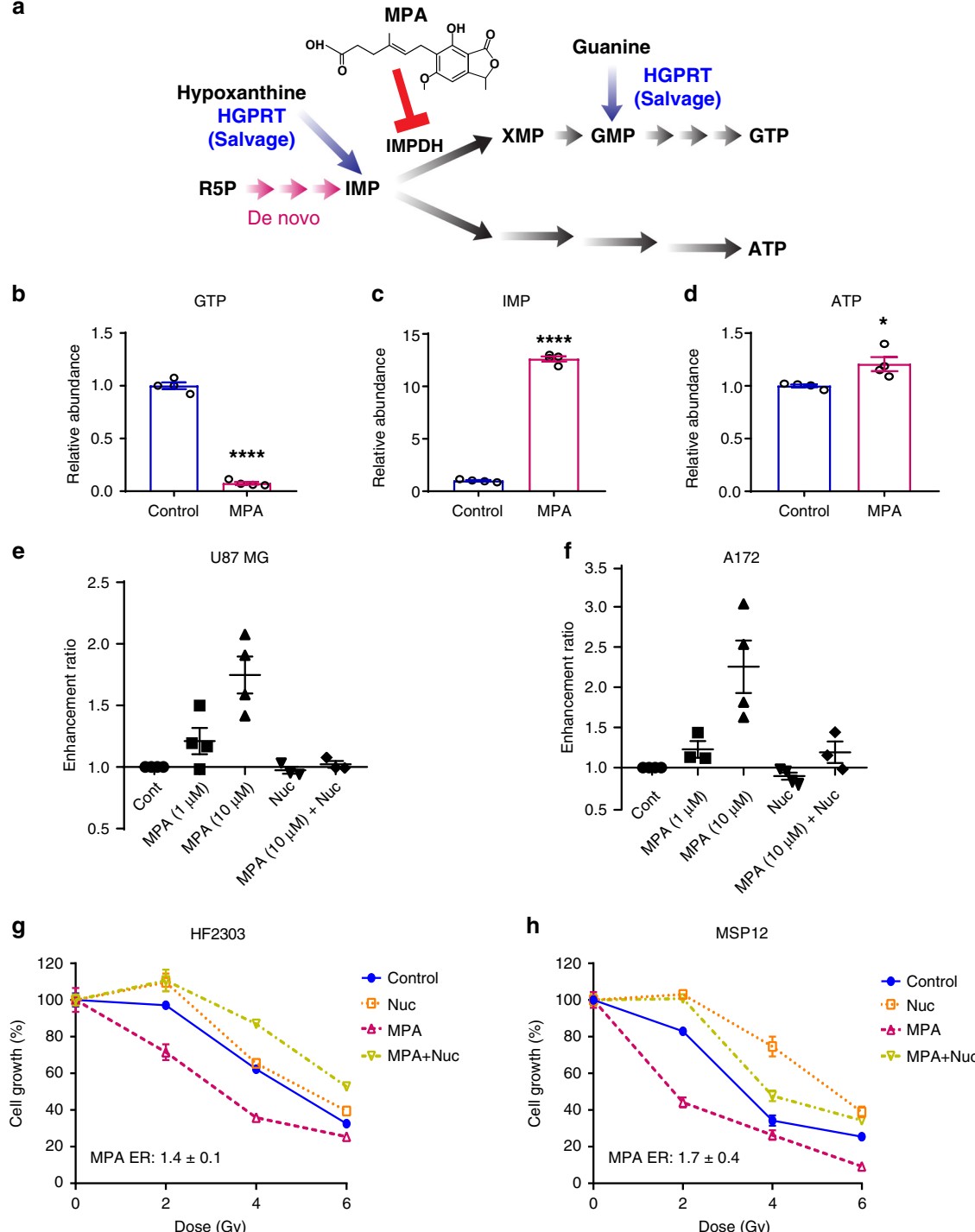

**Fig. 3 Inhibiting GTP synthesis radiosensitizes RT-resistant GBM cell lines and patient-derived neurospheres. a** Purine synthesis schematic. R5P: Ribose 5-phosphate; IMP: inosine monophosphate; GTP: Guanosine-5′-triphosphate; ATP: Adenosine triphosphate; IMPDH: Inosine-5′-monophosphate dehydrogenase; HGPRT: hypoxanthine-guanine phosphoribosyltransferase; MPA: mycophenolic acid. **b–d** U87 MG cells were treated with MPA (10 μM) for 24 h, and then harvested and analyzed by targeted LC-MS/MS. Data are presented as mean ± SEM from four biologic replicates. *p* values are <0.0001, <0.0001, and 0.0237 for Fig. **b–d**, respectively. *$p < 0.05$, and ****$p < 0.0001$ compared with control. The *p* values indicated were obtained by two-tailed unpaired student's *t* test. **e**, **f** After treatment with indicated conditions, cells were replated for colonogenic assay and colonies were stained and counted 10 to 14 days later. Data are presented as mean ± SEM from 4 separate experiments. **g**, **h** HF2303 or MSP12 neurospheres were treated as the timeline shown in Supplementary Fig. 3e. In brief, cells were treated with nucleosides or MPA (10 μM), and retreated with nucleosides 2 h before RT. Cells were replated to 96-well plates (2000 cells/well) 24 h post-RT and cell viability were detected by the Celltiter-Glo kit ~7 days after replating. Fig. **g** and **h** are representative figures from three biologically independent experiments. Error bars show mean ± SEM from representative experiments, which were performed in five (**g**) or six (**h**) technical replicates. ER (mean ± SEM) of MPA from biologic replicates is shown on the lower left of each graph and is calculated as the GI50 of the control-treated cells divided by the GI50 of the MPA-treated cells. Source data are provided as a Source Data file.

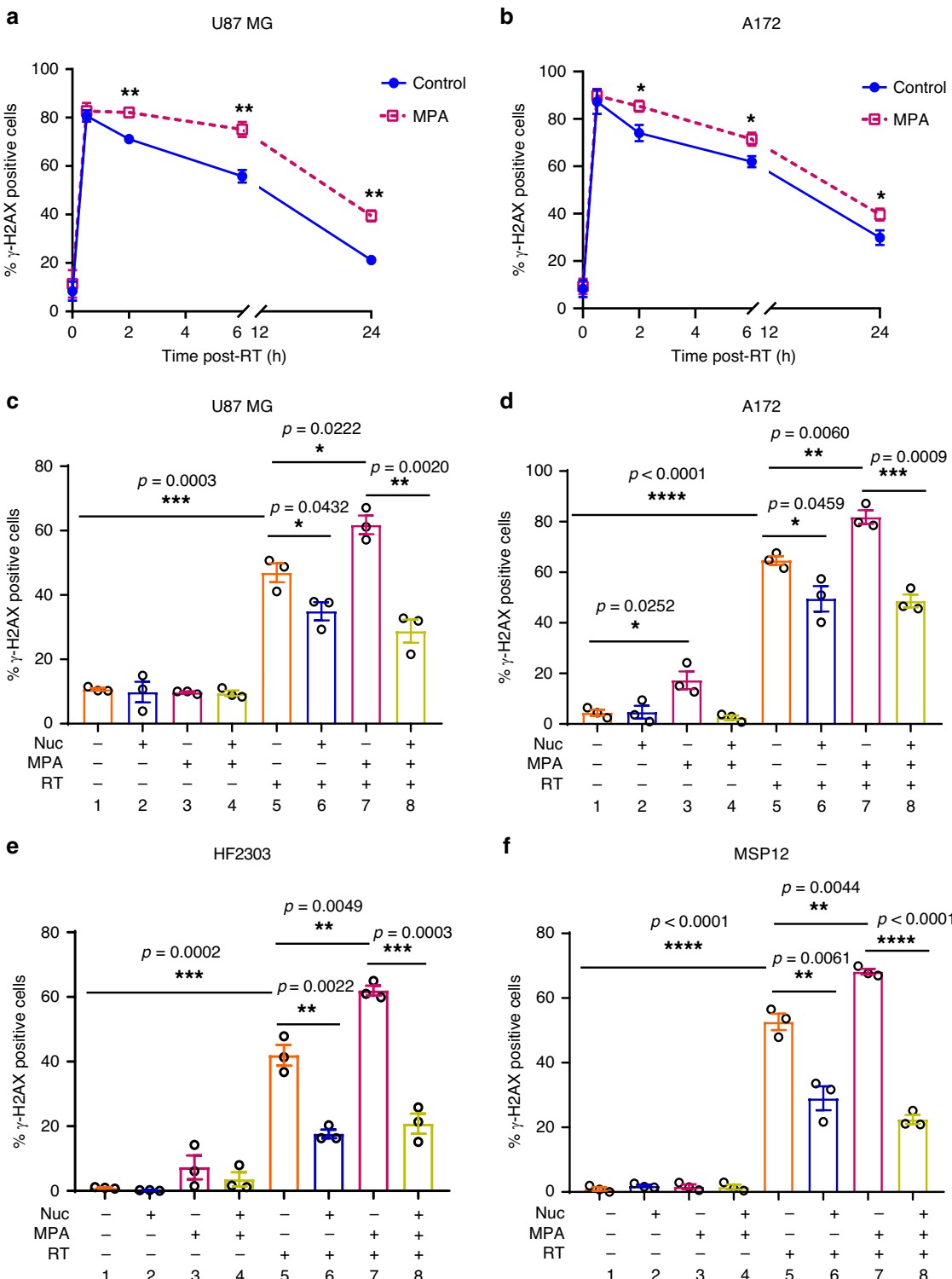

**Fig. 4 Inhibiting GTP synthesis impairs DNA repair in a nucleoside-dependent fashion. a–b** Cells were treated with 10 μM MPA for 24 h and then irradiated with 4 Gy, and cells were harvested at indicated time point for γ-H2AX foci staining. Data are presented as mean ± SEM from three (**a**) or six (**b**) biologic replicates. *p* values of 2, 6, and 24 h are 0.0074, 0.0088, and 0.0036 for Fig. **a**; 0.0220, 0.0273, and 0.0355 for Fig. **b**. RT-resistant cells or neurospheres were treated with MPA (10 μM) and collected for IF γ-H2AX foci staining 6 h post-RT. Data are presented as mean ± SEM from three biologically independent experiments in Fig. **c**–**f**. Fig. **a**–**f**, *p < 0.05; **p < 0.01; ***p < 0.001; ****p < 0.0001. The *p* values indicated were obtained by two-tailed unpaired student's *t* test. Source data are provided as a Source Data file.

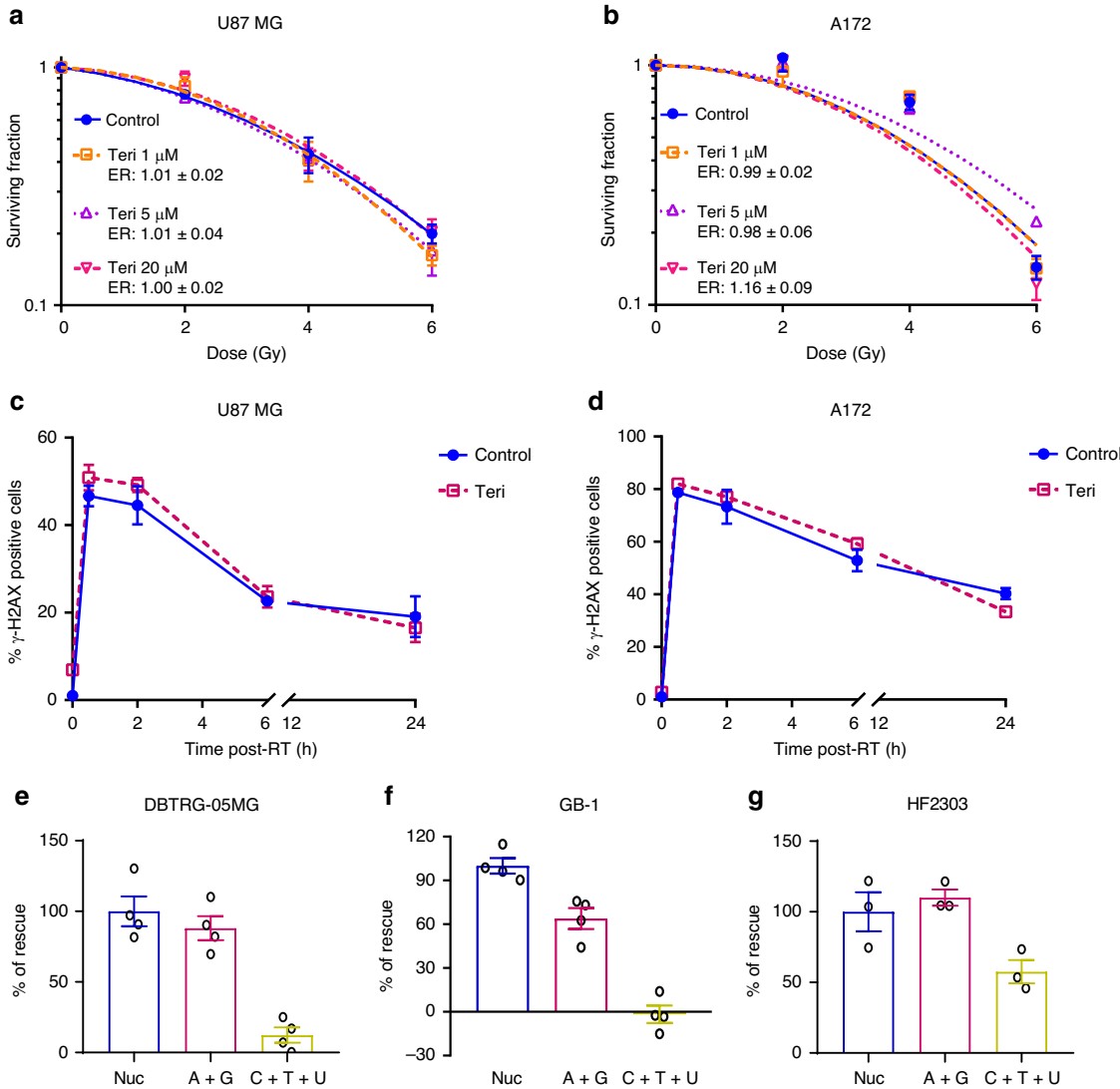

**Fig. 5 Modulating pyrimidine pools has minimal effects on GBM DNA repair and RT-resistance. a, b** U87 MG and A172 cells were treated with varying doses of teriflunomide (Teri) for 24 h, and then irradiated. Cells were replated for colonogenic assay 24 h post-RT. Individual plots show data from one of three biologic replicates and error bars indicate mean ± SEM of technical replicates from that experiment. ER show the mean ± SEM from all three biologic replicates. **c, d** Cells were treated with 20 µM teriflunomide for 24 h and then irradiated with 4 Gy, and cells were harvested at indicated time point post-RT for γ-H2AX foci staining. Data points are averages (mean ± SEM) from three biologic replicates. Cells were treated with pooled nucleosides (Nuc; 8×), or a combination of Adenosine + Guanosine (A + G), or Cytidine + Thymidine + Uridine (C + T + U) for 24 h, and retreated with indicated nucleosides 2 h before RT (4 Gy), followed by IF γ-H2AX foci staining 6 h post-RT. For Fig. **e–g**, data are presented as mean ± SEM from four (**e, f**) or three (**g**) biologically independent experiments. Source data are provided as a Source Data file.

administration of exogenous nucleosides ($p < 0.01$ in U87 MG and $p < 0.001$ in A172, column 7 vs. 8; Fig. 4c, d; Supplementary Fig. 4C, D). Similarly, MPA increased γ-H2AX foci in primary neurospheres ($p < 0.01$ for HF2303 and $p < 0.01$ for MSP12, column 5 vs. 7), which was reversed by nucleoside treatment ($p < 0.001$ forHF2303, $p < 0.0001$ for MSP12; column 7 vs. 8; Fig. 4e, f; Supplementary Fig. 4E, F). Hence, inhibition of de novo purine synthesis impairs DNA repair and radiosensitizes GBM in a nucleoside-dependent fashion in both RT-resistant GBM cell lines and primary patient-derived GBM neurospheres.

**Purines, not pyrimidines, govern RT-resistance in GBM.** To understand more precisely which nucleotides were mediating RT-resistance and DNA repair in GBM, we used teriflunomide[42,43], an FDA-approved inhibitor of dihydroorotate dehydrogenase (DHODH), a rate limiting enzyme in de novo pyrimidine

synthesis (Supplementary Fig. 5A). Teriflunomide decreased pyrimidines concentrations in GBM cells (Supplementary Fig. 5B–F), but did not radiosensitize RT-resistant GBM cell lines (Fig. 5a, b), or impair the ability of these cell lines to repair RT-induced DSBs as measured by γ-H2AX foci (Fig. 5c, d; Supplementary Fig. 5G, H).

We then asked whether the pyrimidine or purine components were responsible for the ability of pooled nucleosides to protect GBM from RT (Fig. 2). In the RT-sensitive DBTRG-05MG cell line, purines alone (adenosine and guanosine) promoted the repair of RT-induced DSBs nearly as much (~90%) as pooled nucleosides (Fig. 5e and Supplementary Fig. 5I). Similar results were seen in the RT-sensitive GB-1 cell line, where purines promoted DSB repair by ~64% compared with pooled nucleosides (Fig. 5f and Supplementary Fig. 5J). Pyrimidines alone (cytidine, uridine, and thymidine), did not promote the repair of RT-induced DSBs in either DBTRG-05MG or GB-1 cells (Fig. 5e,

f; Supplementary Fig. 5I, J). In HF2303 neurospheres, purine nucleosides stimulated DNA repair as much as (~110%) pooled nucleosides, while pyrimidines alone had a more modest effect (Fig. 5g and Supplementary 5K). Thus, purines appear to play a greater role in mediating DNA repair and RT-resistance in GBM than do pyrimidines.

**Inhibiting GTP synthesis radiosensitizes flank models of GBM.** We next sought to understand whether inhibiting GTP synthesis could overcome GBM RT-resistance in tumor models in vivo. We utilized MMF, which is an orally bioavailable pro-drug of MPA that is FDA-approved to treat organ rejection. Because the U87 cell line was one of the most RT-resistant models in our initial profiling (Fig. 1a), we initially used this model in vivo. Once flank xenografts were 80–100 mm³ in size, they were randomized to receive vehicle control, RT alone, MMF or MMF + RT (Supplementary Fig. 6A). We analyzed a subset of tumors 2 h after their second RT dose. Numerous guanylates increased shortly following RT, and this increase was abrogated when RT was combined with MMF (Supplementary Fig. 6B). γ-H2AX levels also increased post-RT, and increased further when RT was combined with MMF (Supplementary Fig. 6C). MMF by itself induced little γ-H2AX. The remaining mice continued with treatment to assess tumor growth. Tumor growth was modestly slowed by treatment with RT or MMF alone, but nearly arrested by the combination of RT and MMF (Supplementary Fig. 6D). These changes were also significant when analyzed as time to tumor tripling ($p < 0.05$, MMF + RT vs. RT; $p < 0.05$, MMR + RT vs. MMF; $p < 0.0001$, MMF + RT vs. Control; Supplementary Fig. 6E).

To extend these findings into models more representative of GBM biology in patients, we performed similar experiments using flank tumors established from HF2303 and MSP12 neurospheres (Fig. 6a). Consistent with the immortalized xenograft model, multiple guanylates were elevated in HF2303 xenografts harvested 2 h after receiving their second RT dose. This increase was again abrogated when MMF was administered along with RT (Fig. 6b). MMF increased RT-induced γ-H2AX in both HF2303 and MSP12 tumors (Fig. 6c). In animals continuing treatment to assess tumor response, single agent MMF and RT modestly slowed both HF2303 and MSP12 tumor growth. However, combined MMF and RT significantly slowed tumor growth (Fig. 6d, e), and increased the time to tumor tripling (Fig. 6f, g). Median days to tumor tripling are 12 (Control), 16 (MMF), 23 (RT), and 33 (MMF + RT) for HF2303 and 10 (Control), 12 (MMF), 13.5 (RT), and 19 (MMF + RT) for MSP12, respectively. The effects of treatments on normal tissues were minimal, as reflected by relatively unchanged body weight during drug treatment (Supplementary Fig. 6F, G). Consistent with the observed treatment efficacy, combined MMF and RT decreased the expression of the cell proliferation marker Ki-67 compared with either treatment in isolation in all three models (Supplementary Fig. 6H).

**MMF augments RT efficacy in an orthotopic PDX model of GBM.** Because the GBM microenvironment and poor intracranial exposure can limit drug efficacy, we sought to understand whether MMF would be efficacious in an intracranial GBM model. We engineered luciferase expression into GBM38, which is an intrinsically RT-resistant PDX model of primary GBM from the Mayo Clinic Brain Tumor Patient-Derived Xenograft National Resource[44]. After intracranial surgeries, tumor initiation was confirmed by bioluminescence imaging and tumor-bearing animals were randomized to receive control, RT, MMF, or MMF + RT + MMF (Fig. 7a). Tumor bioluminescence was profoundly decreased by combined RT and MMF treatment, but only slightly

affected by MMF or RT when given alone (Fig. 7b, c). Consistent with these observations, combined MMF and RT prolonged mouse survival (median 62 days) compared with control treatment (median 42 days), while MMF (median 43 days) or RT (median 45.5 days) alone had minimal effects (Fig. 7d). Hence, inhibition of GTP synthesis with MMF can overcome GBM RT-resistance intracranially.

**High *IMPDH1* is associated with inferior GBM patient survival.** Finally, we asked if these data were reflected in the outcomes of patients with GBM. We identified 208 patients from the Cancer Genome Atlas Pan-Cancer Atlas with newly diagnosed primary IDH wild type gliomas (so-called "molecular GBMs") whose samples had passed the Pan-Cancer quality assurance, the vast majority of whom received RT[45]. Increased transcript expression (>median) of *IMPDH1*, the rate limiting enzyme in de novo GTP synthesis and target of MPA/MMF, was associated with inferior overall survival (HR 0.60, 95% CI 0.43–0.82, $p = 0.002$). The *IMPDH1* high and low groups were similar with respect to known clinical and pathologic determinants of survival including age (median 60 vs. 59 years) and *MGMT* promoter methlation (39 vs. 40%). We could not quantify the extent of resection in *IMPDH1* high and low groups and therefore do not know if this key variable is partially responsible for the apparent decreased survival in patients with low *IMPDH1* expression[46]. Increased expression of the rate limiting enzymes in de novo ATP synthesis (*ADSS*: adenylosuccinate synthase and *ADSL*: adenylosuccinate lyase) or de novo pyrimidine synthesis (*DHODH*: dihydroorotate dehydrogenase and *CAD*: carbamoyl-phosphate synthetase 2, aspartate transcarbamylase, and dihydroorotase) were not associated with decreased survival in patients with newly diagnosed IDH wild type glioma (Fig. 7e).

We performed an exploratory analysis of the small number ($n = 22$) of patients in the PANCAN dataset with IDH mutant GBM. Neither *IMPDH1* nor *IMPDH2* were prognostic in these patients (Supplementary Fig. 7A). Intriguingly, high expression of *DHODH*, a rate-limiting step in de novo pyrimidine synthesis, was adversely prognostic in patients with IDH mutant GBM, which suggests that further investigation of pyrimidine biosynthesis could be interesting in gliomas with an IDH mutation. Together, these in vitro, in vivo and patient-level data suggest that purines, especially GTP, mediate RT-resistance, and DNA repair in IDH wild type GBM and that inhibition of GTP synthesis could be a promising therapeutic strategy for GBM, especially when combined with RT.

## Discussion
Intratumoral genomic heterogeneity in GBM has limited the efficacy of personalized targeted therapies. To overcome this barrier, we sought to discover metabolic pathways that caused RT-resistance in GBM independently of genotype. By analyzing how intracellular metabolites correlated with GBM RT-resistance, we found that low levels of nucleobase-containing metabolites were strongly associated with sensitivity to RT. This association was causal, as supplementing GBM cells with exogenous nucleosides protected them from RT by promoting the repair of RT-induced DSBs. The protective effects of these nucleosides were primarily due to purines rather than pyrimidines. This relationship between nucleotide pools and RT-resistance has important therapeutic applications. Depleting intracellular GTP pools with FDA-approved drugs sensitized GBM cell lines to RT by slowing the repair of dsDNA breaks. Inhibiting salvage GTP synthesis or depleting pyrimidine pools had no such effects. These results were recapitulated in neurosphere, flank-xenograft, and intracranial PDX models of GBM. In addition, high

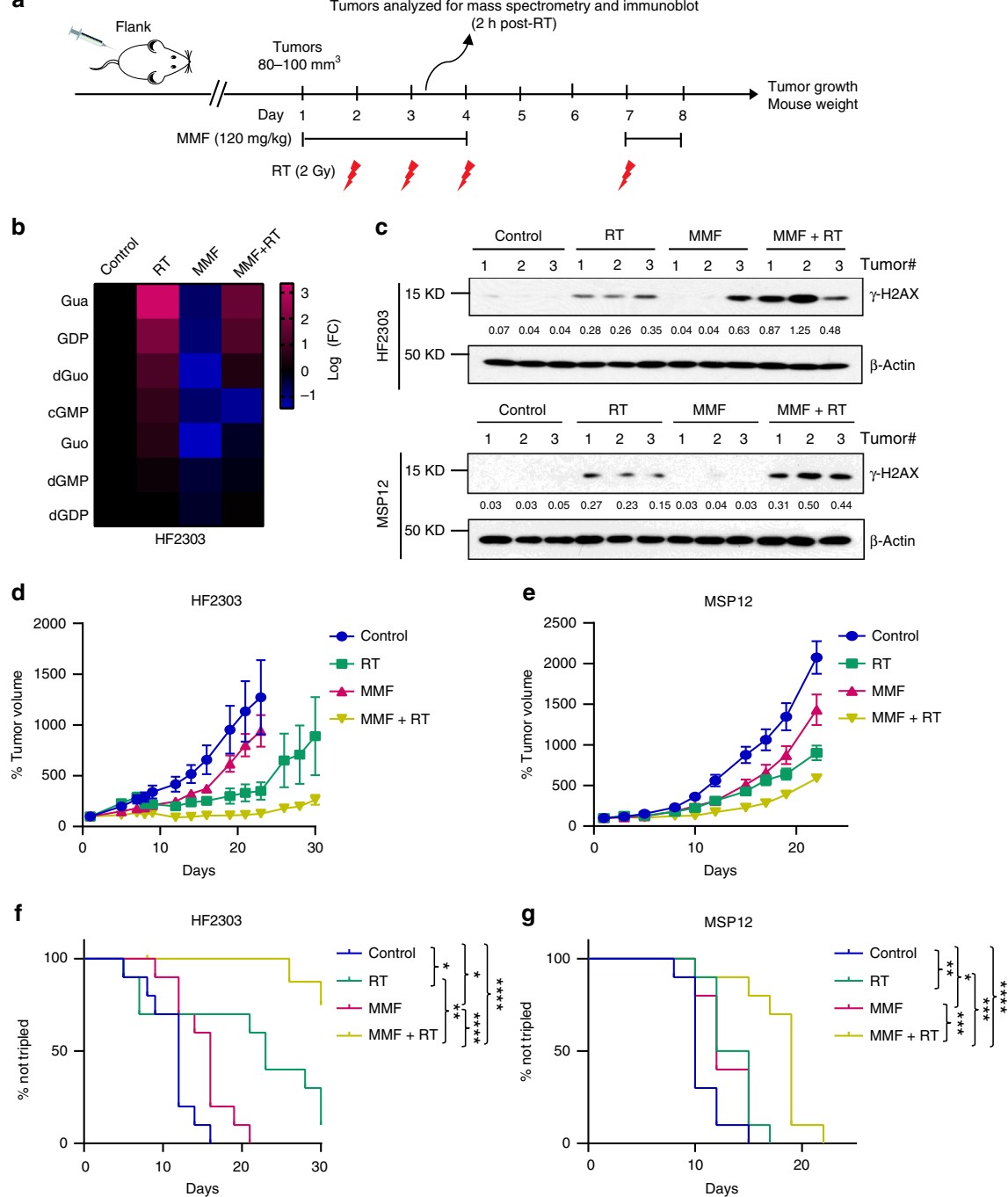

**Fig. 6 MMF augments RT efficacy against immortalized GBM xenografts. a** A schematic timeline of HF2303 and MSP12 flank models. HF2303 and MSP12 xenograft GBM models were established and randomized as described in "Methods". Mycophenolate mofetil (MMF) (120 mg/kg) was administered via oral gavage once daily beginning the day prior to RT and ending the day after. MMF was given 2 h before RT and held on weekends (six total doses). RT (2 Gy/fraction) was administered over four daily fractions on weekdays. **b, c** A subset of tumors harvested 2 h after receiving the second RT dose were analyzed by LC-MS/MS (**b**), or were ground and lysed for immunoblotting assay with indicated antibodies. The bands were quantified using Image J 2.0 software and the quantified numbers were labeled under each band (**c**). Fig. **c** are representative figures from three biologically independent experiments. **d, e** Tumor volumes for the indicated treatment groups are normalized to the individual tumor sizes defined on day 1. Error bars indicate mean ± SEM from ten tumors of 5 mice per group. **f, g** Kaplan–Meier estimates of time to tumor tripling. *p* values for Control vs. RT are 0.0117 Fig. **f** and 0.0089 Fig. **g**; Control vs. MMF are 0.0133 Fig. **f** and 0.0324 Fig. **g**; Control vs. MMF + RT are <0.0001 Fig. **f, g**; RT vs. MMF + RT are 0.0026 Fig. **f** and 0.0009 Fig. **g**; MMF vs. MMF + RT are <0.0001 Fig. **f** and 0.0006 Fig. **g**; *\*p* < 0.05; *\*\*p* < 0.01; *\*\*\*p* < 0.001; *\*\*\*\*p* < 0.0001. The *p* values were obtained by the log-rank (Mantel–Cox) test. Source data are provided as a Source Data file.

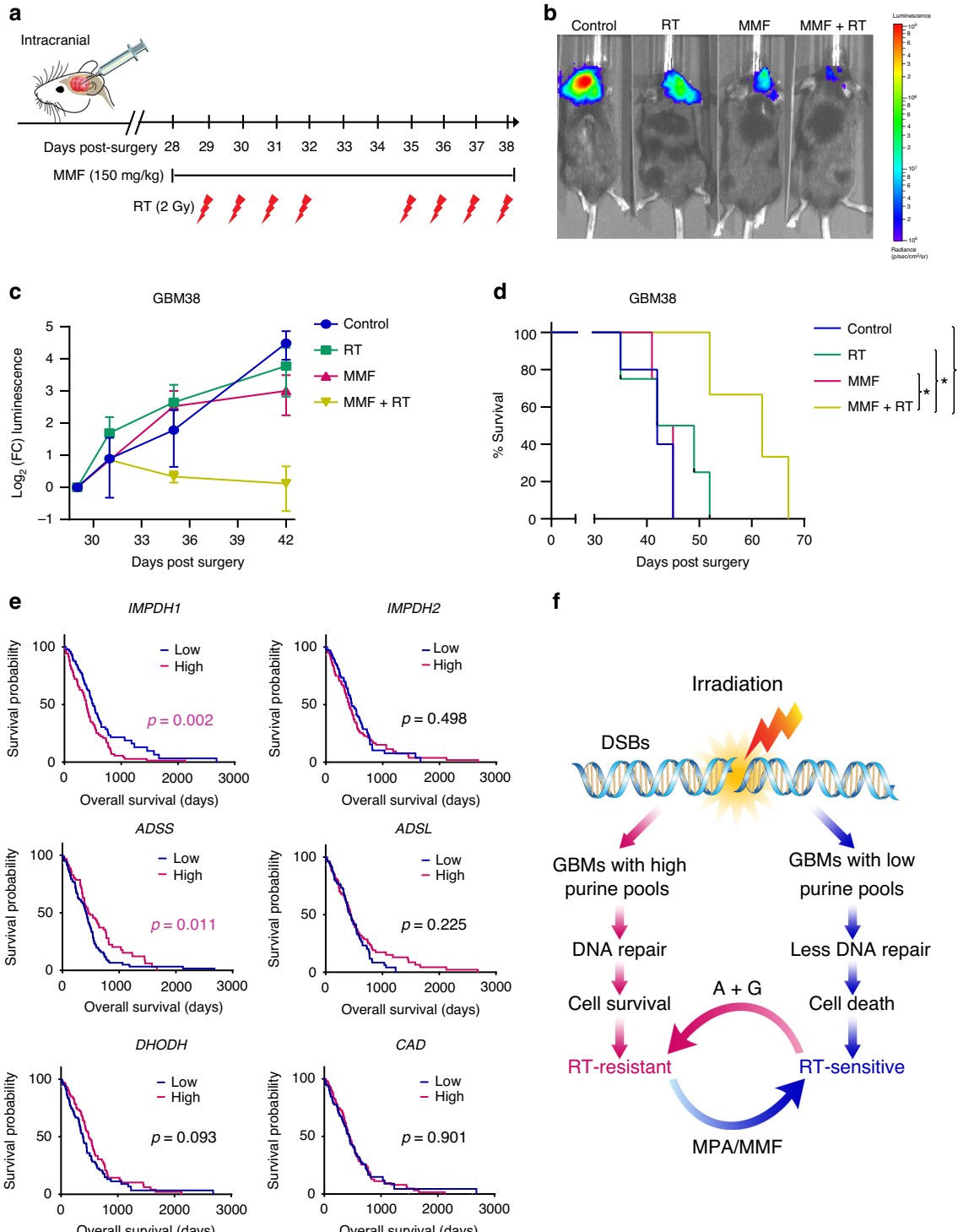

**Fig. 7 Purine metabolism mediates RT resistance in an orthotopic patient-derived GBM model. a** Luciferase-positive RT-resistant GBM38 patient-derived xenograft cells were orthotopically implanted and, 28 days later, brain tumor-bearing mice were randomized to treatment with drug vehicle, RT, MMF, or MMF + RT (3–5 animals per group). MMF was administrated from Day 28 to 38 (11 doses), and RT was given from Day 29 to 32 and Day 35 to 38, respectively. **b** Mice were treated with 150 mg/kg D-luciferin and imaged 10 min post-injection. **c** Total flux of equal-area ROIs at each time point were normalized to flux at the first day of treatment and used to approximate tumor progression. Data are presented as mean ± SEM from three (MMF + RT) or four (Control, RT, MMF) independent mice per group. **d** Kaplan–Meier survival curve. Mice were monitored daily and euthanized when they developed neurologic symptoms. *p* value for RT vs. MMF + RT is 0.0409; 0.0213 for MMF vs. MMF + RT; 0.0151 for Control vs. MMF + RT; *p < 0.05. The *p* value was obtained by the log-rank (Mantel–Cox) test. **e** Kaplan–Meier overall survival curve of 208 patients from the Pan-Cancer Atlas with newly diagnosed *IDH* wild type GBM. High and low groups are defined by median expression of key rate limiting enzymes of nucleotide synthesis. The *p* values were obtained using the log-rank (Mantel–Cox) test. **f** Working model. RT induces DSBs in GBMs. High de novo purine synthesis promotes GBM survival by stimulating dsDNA repair, cell survival and recurrence after RT. Supplementing cells with purines (A + G) promotes RT-resistance while inhibiting de novo purine synthesis with mycophenolic acid (MPA) or mycophenolate mofetil (MMF) promotes RT-sensitivity. Source data are provided as a Source Data file.

expression of the rate limiting enzyme in de novo GTP synthesis was associated with inferior survival in IDH wild type GBMs but not in IDH mutant GBMs. In summary, we have found that purine, and especially guanylate, metabolism mediates RT-resistance in GBM and can be targeted with FDA-approved drugs (Fig. 7e).

These results add to a growing body of literature indicating that purine synthesis contributes to the aggressive behavior of GBM and other cancers[47]. High rates of de novo purine and pyrimidine synthesis promote the maintenance and tumorigenic capacity of glioma-initiating cells, which are thought to contribute to therapy resistance and tumor recurrence in GBM[48,49]. Our data suggest that the high rates of de novo, but not salvage, purine synthesis in these tumorigenic cells may contribute to their enhanced ability to repair RT-induced DNA damage and mediate tumor recurrence. De novo purine synthesis can generate both GTP and ATP. De novo GTP synthesis is preferentially upregulated in GBM, while de novo ATP synthesis is similarly active in both normal brain tissue and GBM. This upregulation of GTP synthesis promotes tRNA and rRNA synthesis, nucleolar transformation, and GBM proliferation[50]. This importance of GTP was recapitulated in our studies, as guanylates were most strongly correlated with GBM RT-resistance and inhibiting de novo GTP synthesis alone was sufficient to overcome GBM RT-resistance.

These findings suggest that MMF should be evaluated for therapeutic benefit in patients with GBM, particularly in combination with RT. Because MMF is FDA-approved for other indications, the barrier to clinical translation is low. Furthermore, standard dosing of MMF appears to achieve active intracranial concentrations, based on published pharmacokinetic studies[34], our own results in orthotopic GBM tumors, and activity in patients with neurosarcoidosis[35]. Because normal glia and neural stem cells have lower demands for GTP synthesis and may preferentially rely on salvage pathways[50], such a therapeutic strategy may have minimal normal tissue toxicity.

Like the small number of therapies with proven benefit in GBM, inhibitors of purine synthesis do not require a precise oncogenic event for activity. Therefore, these inhibitors may have clinical benefit despite the intratumoral genomic heterogeneity that characterizes GBM. Indeed, many of the heterogeneous oncogenic alterations that drive GBM including mutations, deletions, or amplifications in *PTEN, EGFR,* and *PIK3CA* can cause the similar metabolic phenotype of elevated de novo purine synthesis[18–20,51]. Thus, a genomically heterogenous GBM[7–9] may exhibit a relatively homogeneous metabolic phenotype of elevated de novo purine synthesis, which could be exploited therapeutically to overcome RT-resistance.

Our study raises several questions. How purines, especially GTP, regulate RT-resistance and dsDNA repair in GBM remains to be defined. Because modulating pyrimidine levels did not cause similar effects as modulating purines, we believe that this link is likely due to active signaling, perhaps through a GTP-activated protein, rather than the more simplistic explanation that modulating nucleotide pools alters the availability of the substrates for DNA repair. Our experiments were entirely carried out in models of GBM without mutations in *IDH1/2*, which represent the vast majority of GBMs in patients. Whether the rarer secondary GBMs containing the *IDH1* or *IDH2* mutation exhibit a similar relationship between de novo purine synthesis and therapy resistance remains to be defined.

In summary, we have defined purine, and especially guanylate, metabolism as a mediator of RT-resistance in GBM. These findings have motivated the development of a clinical trial to test whether MMF achieves effective concentrations in GBM tissue in patients and whether it is safe and effective in combination with RT for patients with this disease.

## Methods

**In vivo xenograft models.** All mouse experiments were approved by the University Committee on Use and Care of Animals at the University of Michigan. C.B-17 SCID mice (female, 4–7 weeks old) were obtained from Envigo and maintained in specific pathogen-free conditions. University of Michigan's Unit for Laboratory Animal Medicine (ULAM) ensured that the housing temperature was kept at 74 °F, relative humidity between 30 and 70%, and was on a light/dark cycle of 12 h on/12 h off. Cells were resuspended in 1:1 PBS:Matrigel (BD Biosciences) and subcutaneously injected into the bilateral dorsal flanks of mice. Once the tumor volume reached ~80–100 mm³, mice we randomized into four arms, including vehicle control (0.5 (w/v) methylcellulose/0.1% (v/v) Polysorbate 80), MMF alone, RT alone, or combined RT and MMF. MMF (120 mg/kg) and/or RT (2 Gy/fraction) were administered as shown in Fig. 6a and Supplementary Fig. 6A. A subset of tumors analyzed for biologic endpoints (including mass spectrometry and immunoblotting) was taken 2 h after their second RT (or sham RT) dose. The remainder of the animals continued with treatment and had tumor volume and body weight measured three times weekly. Tumor volumes were determined using digital calipers and the formula $(\pi/6)$ (Length × Width²).

**Orthotopic patient-derived xenograft model.** RT-resistant GBM38 PDXs[44] were obtained from Dr. Jann Sarkaria and propagated as subcutaneous flank tumors in female CB17-SCID mice. After short-term explant culture, cells were infected with lentiviruses harboring fluc (lenti-LEGO-Ig2-fluc-IRES-GFP-VSVG) and enriched for GFP-positive populations by FACS, followed by subcutaneous injection into the flanks of mice to propagate GBM tissue. GBM cells expressing luciferase were then isolated from the PDX tumors for orthotopic implantation in 6-week-old male and female Rag1-KO mice (B6.129s7-RAG1 tm/Mom/J) obtained from the Jackson Laboratory. After mouse anesthetization, a small scalp incision (<1 cm) was made, and a burr hole was produced at coordinates of 1 mm forward and 2 mm lateral from the bregma. Cells (~5 × 10⁵) were then injected at a depth of 3 mm at a rate of 1 μL/min. Carprofen (5 mg/kg) was subcutaneously administered daily for 48 h following procedure.

Treatment regimens for randomized brain tumor-bearing mice were initiated 4 weeks post implantation as shown in Fig. 7a. To assess the bioluminescence of brain tumors, mice were treated with 150 mg/kg D-luciferin by intraperitoneal injection and then imaged 10 min later under anesthesia (2% isoflurane inhalation) using an IVIS™ Spectrum imaging system (PerkinElmer). Bioluminescence values for mice euthanized midway through bioluminescence experiments were imputed using most recent measurement prior to euthanization.

**Mass spectrometry analysis.** GBM cells or tumor samples were mixed or homogenized in cold (−80 °C) 80% methanol. After centrifuging, samples were normalized and lyophilized by speed vac. Dried pellets were resuspended in 1:1 methanol:H₂O before LC-MS/MS analysis. Samples were run in triplicate on an Agilent QQQ 6470 LC-MS/MS with ion pairing chromatography acquiring dynamic multiple reaction monitoring for 226 metabolites with a delta retention time (RT) window of 1 min. Data were preprocessed by applying a threshold area of 3000 ion counts and a coefficient of variation of 0.5 among triplicates. All chromatography analysis was done with Agilent MassHunter Quantitative Analysis 9.0.647.0. Raw data from these studies are included as a supplementary file (Supplementary Data 2).

Initial cell metabolic profiling shown in Fig. 1c–e was performed by Metabolon, Inc. Briefly, the 23 GBM cell lines were cultured in DMEM to 75–90% confluence and harvested 1 h after addition of fresh media (Fig. 1c). Two RT-resistant cell lines (U87 MG and A172) and two RT-sensitive cell lines (KS-1 and U118 MG, Fig. 1d, e) were irradiated (8 Gy) and harvested after an additional 2 h incubation. The global metabolic profiles of all the harvested cell samples (4–5 biologic replicates per condition) were determined by Metabolon. Data from this profiling effort are attached as a separate supplementary file (Supplementary Data 1).

¹⁵N-amide glutamine tracing study shown in Supplementary Fig. 3A–D was performed at the University of Michigan Metabolomics Core. U87 MG cells labeled with 2 mM ¹⁵N-amide glutamine were mixed in a solution of methanol, chloroform, and water (8:1:1) for homogenizing and metabolite extraction. After centrifuging, 100 μL of the extraction solvent was transferred to an autosampler vial for LC-MS analysis. Nucleotide/Deoxynucleotide analysis was performed on an Agilent system consisting of an Infinity Lab II UPLC coupled with a 6470 Triple Quad (QqQ) mass spectrometer (Agilent Technologies, Santa Clara, CA.). Metabolites were identified by matching the RT and mass (±10 ppm) to authentic standards. Isotope peak areas were integrated using MassHunter Quantitative Analysis vB.09.00 (Agilent Technologies, Santa Clara, CA)[52].

**Metabolic pathway analysis.** The normalized metabolite intensity levels were z-transformed (i.e., zero mean and unit variance across all cell lines) to enable comparison on the same scale[53]. Metabolites with z-score below −1 or above +1 were assumed to be downregulated or upregulated. Metabolites were then grouped into corresponding pathways based on annotation from Metabolon. A pathway-level up or downregulation score was determined by calculating the ratio of the total number of metabolites that were significantly up- or down-regulated to the total metabolites measured in that pathway. This was then correlated with RT-

resistance score using Pearson's linear correlation function in MATLAB. The pathways with significant correlation ($p < 0.05$) were also found to be significant after Benjamin–Hochberg false discovery rate correction (FDR < 0.1). Significantly correlated pathways were then visualized on a human metabolic network map (Supplementary Fig. 1H) using the iPath pathway explorer[54].

**TCGA clinical and molecular data**. TCGA Pan-Cancer Atlas LGG and GBM cohorts were used for survival analysis and gene expression profiling[55]. For the purposes of the current study, curation of these cases was performed to include only *IDH* wild type primary/untreated samples, WHO grades II–IV. We further excluded cases based on those that were masked ("Do_not_use") according to the Pan-Cancer Atlas sample quality annotations (http://api.gdc.cancer.gov/data/1a7d7be8-675d-4e60-a105-19d4121bdebf). From the initial 1118 cases identified in the LGG and GBM project, 208 *IDH* wild type primary tumors were used for further analyses. Of the 208 *IDH* wild type TCGA cases we used for survival analyses, there were 137 patients from the "Glioblastoma Multiforme" study (and thus were grade IV by histologic criteria). A total of 71 patients were identified from the LGG dataset who had either or both of TERT promoter mutation, the +7/−10 signature, and EGFR amplification. If one of these are present, it now satisfies the criteria for Glioblastoma, WHO grade IV[56]. To further differentiate between IDH wild type and mutant GBM, we included 22 patients with IDH mutant gliomas, which are either grade 4 by histology or by the presence of *CDKN2A/B* homozygous deletion.

Gene expression data (RNA-seq) from the LGG and GBM cohorts was downloaded from cBioPortal (http://www.cbioportal.org/, accessed 9/29/2019). RSEM-normalized expression values were then stratified into low and high expressing groups using a median cutoff. Overall survival was used as the clinical endpoint and survival analytics were obtained from the TCGA Pan-Cancer Clinical Data Resource (TCGA-CDR). Survival curves were generated using the Kaplan–Meier method and significance assessed using the log rank test. Statistical significance was set at $p < 0.05$. Anonymized clinical data was obtained from the above publicly available sources. Thus, neither informed consent nor IRB approval was sought.

**Statistical methods**. Clonogenic survival, γ-H2AX foci formation, comet assay, Ki-67 IHC staining, and metabolite level analysis after MPA and teriflunomide treatment were analyzed by unpaired two-tailed *t*-tests using GraphPad Prism Version 8 with the Holm-Sidak method employed to account for multiple comparisons when appropriate. Tumor volume of GBM xenografts was normalized to 100% at the first day for each group. Time to tumor tripling in each group was determined by identifying the earliest day on which it was at least three times as large as on the first day of treatment and then estimated by the Kaplan–Meier method and compared using the log-rank test. Significance threshold was set at $p < 0.05$.

**Cell lines and additional methods**. Detailed source of GBM cell lines is outlined in Supplementary Table 1. Additional methodological detail, including a description of cell culture and reagents, clonogenic survival assay, CCLE analysis, gene knockdown, immunofluorescence, flow cytometry, long-term neurosphere assay, alkaline comet assay, immunoblotting, and immunohistochemistry accompanies this manuscript in the Supplementary Methods.

**Reporting summary**. Further information on research design is available in the Nature Research Reporting Summary linked to this article.

## Data availability

Source data underlying Figs. 1a–e, 2b–i, 3b–h, 4a–f, 5a–g, 6b–g, 7c–e and Supplementary Figs. 1A, B, F, G, 3A–D, F–H, 5B–F, 6B–H, and 7A are provided as a Source Data file. The metabolomic data of Fig. 1c–e (Supplementary Data 1), and Fig. 3b–d, Fig. 6b and Supplementary Fig. 5B–F, 6B (Supplementary Data 2) for this article are available as Supplementary Information files. Uncropped images for immunoblots are shown in Supplementary Figs. 8. Flow cytometry gating information is shown in Supplementary Figs. 9 and 10. Further data are available from the corresponding author upon reasonable request. Source data are provided with this paper.

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

## Acknowledgements

We thank Rob Mohney, Ph.D., Ed Karoly Ph.D. and colleagues at Metabolon Inc for their metabolomics profiling of GBM cell lines. We thank Steven Krongenberg for his assistance with illustrations. D.R.W. was supported by grants from the American Cancer Society, the Forbes Institute for Cancer Discovery, the NCI (K08CA234416) and the Jones Family Foundation Fund within the Chad Carr Pediatric Brain Tumor Center. W. Z. was supported by the Postdoctoral Translational Scholar Program (UL1TR002240) from the Michigan Institute for Clinical & Health Research of University of Michigan. A. J.S. was supported by the Rogel Cancer Center Post-doctoral Research Fellowship (G023496). B.S.N. was supported by T32-DK094775 and T32-CA009676. C.J.H. was supported by K99CA241357, P30DK034933 and F32CA228328. C.A.L. was supported by a 2017 AACR NextGen Grant for Transformative Cancer Research (17-20-01-LYSS) and an ACS Research Scholar Grant (RSG-18-186-01). Metabolomics studies performed at the University of Michigan were supported by NIH grant DK097153, the Charles Woodson Research Fund, and the UM Pediatric Brain Tumor Initiative. D.R.W. and C. A.L. received support from UMCCC Core Grant (P30CA046592), which also supported the Rogel Cancer Center Preclinical Imaging & Computational Analysis Shared Resource & the Experimental Irradiation Shared Resource. M.A.M. is supported by R01CA240515 and R01CA156744. T.S.L. was supported by NIH grant U01CA216440 and UMCCC Core Grant P30CA46592.

## Author contributions

D.R.W., and W.Z. conceptualized and designed the study. W.Z., Y.Y., J.J.D., A.J.S., K.W. R., and D.R.W. developed the methods and performed the experiments. L.Z., B.S.N., C.J. H., A.W., M.K., C.A.L., and D.R.W. completed the data acquisition of LC/MS. P.S., S.G. Z., F.G., S.C., D.P., W.Z., Y.Y., and D.R.W. analyzed and interpreted the data. W.Z., Y.Y., C.A.L., T.S.L., A.J.S., M.G.C., P.L., and D.R.W. wrote, reviewed and revised the paper. M. D., M.A.M., C.K.W., H.S., A.R., Y.U., J.N.S., J.X., M.G.C., and P.L. provided administrative, technical and material supports. C.A.L. and D.R.W. supervised the study.

## Competing interests

D.R.W. has received research grant support from Innocrin Pharmaceuticals Inc. and Agios Pharmaceuticals Inc. for work unrelated to the content of this manuscript. The other authors have declared that no conflict of interest exists.
