## [Peer Review File · Nature Communications]

Reviewers' comments:

Reviewer #1 (Remarks to the Author); expert on GBM, mouse models and radioresistance:

As an approach to overcoming the presumed intra-tumor heterogeneity in radiosensitivity existing in GBMs, this manuscript investigates purine synthesis as a determinant of GBM radioresistance. Towards this end, a panel of 23 established GBM cell lines are ranked according to their in vitro radiosensitivity as determined by clonogenic assay and subjected to metabolic analysis, which suggested that purines were related to GBM radioresistance. Inhibition of purine synthesis using mycophenolic acid (MPA) was shown to enhance radiosensitivity and replacement with exogenous nucleotides was shown to abrogate the radiosensitization. Similar results were obtained using 2 patient-derived GBM neurosphere cultures. In vivo experiments were then performed using flank xenografts grown from a GBM cell line and the neurospheres. Based on the data presented, the authors concluded that purine synthesis mediates the radioresistance of GBM and propose a clinical trial combining an MPA derivative and radiotherapy. However, there are a number of experimental deficiencies in this study (see below). Moreover, the relevance of the established cell lines to the biology/radiobiology of GBMs can be questioned. While the in vitro studies are aided by the addition of the 2 neurosphere cultures, all in vivo experiments are performed using flank xenografts, which do not account for the unique circumstances of brain microenvironment. The absence of an orthotopic model system questions the relevance of these results to GBMs and their therapy.

Fig.1. The basis for the ranking of radiosensitivity among the established cell lines (1A) is unclear. It appears that for the majority of lines there is no statistically significant difference in Dmid. Whereas it seems reasonable to focus subsequent experiments on cell lines on the extremes of the measurement, the Dmid ranking does not appear to correspond to the actual survival curves shown for "sensitive" lines in figure 5. That is, U118MG, DBTRG-05MG and GB-1 are classified as sensitive according to Dmid, yet 6Gy results in a surviving fraction of 0.2 in U118MG (similar to U87 and A172, which are classified as resistant – figure 5), and almost 0.01 in the other 2 lines. In figure 1D, it is stated that 2 resistant and 2 sensitive lines were subjected to metabolic evaluation after irradiation, yet only 2 columns were presented. Were the values averaged?

Fig.2. The alkaline comet assay is claimed to measure radiation-induced DSBs. This is inaccurate in that this assay primarily measures radiation-induced single strand breaks, which are a nonlethal event. To measure DSBs, it is necessary to use the neutral comet assay. This is significant in that the comet results were used to claim that the nucleosides do not affect the initial level of radiation-induced DSBs, but only influenced their repair. The rationale for performing the alkaline comet assay was that gammaH2AX analysis cannot distinguish between the induction and repair of DSBs, which is also inaccurate. gammaH2AX foci correspond to the initial level of DSBs at 0.5h after irradiation; their dispersal correlates with repair. In 2E-G, nucleosides were shown to decrease the initial level of gammaH2AX foci detected at 0.5h as well as those detected at times out to 24h. Thus, the data presented actually suggest that the nucleoside addition reduces the number of radiation-induced DSBs, which is in conflict with the conclusions derived from the comet assay.

Fig.4. Whether the differences between Control and MPA are statistically significant should be shown.

Fig. 7F. For this analysis GBM is combined with low grade gliomas (LGG). To this point the entire paper had focused on GBM. The biology as well as the treatment of LGG and GBM are different. Moreover, if it was just GBM, the data would have to take into account percent resection, MGMT status and patient age, which are not accounted for in LGG analyses. The combination of these tumor types certainly complicates data interpretation and should be clearly justified.

Reviewer #2 (Remarks to the Author); expert on GBM and resistance:

This is an interesting study, extending the previous work of Jeremy Rich and colleagues regarding

the importance of purine metabolism in glioblastoma stem-like cells (Wang Nat Neurosci 2017), and targeting of pyrimidine synthesis to overcome resistance to drugs (Wang Sci Trans Med 2019).

Here, the authors report that inhibition of purine metabolism can lower radioresistance in glioblastoma cell lines, spheroids, and subcutaneously growing xenograft models in mice - by interfering with DNA repair. This is in principle a valuable extension of the previous work, and of high translational significance. The drug they use (MMF) is approved, widely used in the clinic, and even capable of passing the blood-brain barrier (a fact the authors should discuss!) - which makes it to a very interesting candidate for future combination trials with radiotherapy in the clinic.

The data appears solid, the manuscript is very well written, the statistics appears adequate and sound, and the conclusions well founded.

I have two major issues:

1.) It is well known today that radioresistance in vitro, and very likely also in non-orthotopic tumor models in vivo is a very poor readout for the "real" radioresistance seen in glioblastoma growing where it belongs (brain). There are several important cellular mechanisms of radioresistance that appear to be brain-specific (e.g., see Osswald et al., Nature 2015). Therefore, the authors MUST add data of patient-derived glioma stem-like cell lines (NOT U87, which is a very, very bad cell line for glioma research today), growing in the mouse brain - and confirm their radiosensitization findings with MMF. As discussed above, they are in a very good position: MMF is brain-penetrant, which is a rare feature for oncological drugs, so these are highly meaningful experiments that, if positive, would dramatically increase the value of this manuscript.

2.) The supportive patient data (Fig. 7F) is interesting. I would like to see the same data for IDH-mutant gliomas. Is this mechanism of (potential) relevance here, too? - I am aware that IDHmut glioma models in vitro and in vivo are tricky, so I am not asking to provide experimental data - but in silico analyses from the TCGA and other databases are straightforward, and would also benefit this manuscript.

Reviewer #3 (Remarks to the Author); expert on organic medicinal chemistry and drug design:

This manuscript systematically demonstrates that radiation (RT) resistance in GBM cells results from an increased ability to repair RT-induced DSBs. Further, increased expression of IDH1 is associated with RT-resistance. The study also demonstrates that GBMs with depleted purines resulted in RT-sensitivity and that protection with added nucleotides in RT-sensitive GBMs facilitates repair of DSB and manifests resistance. In contrast, inhibition of de novo purine synthesis, particularly GTP synthesis via MMF, results in RT-sensitization of GBMs in cells as well as in patient-derived GBN neurospheres.

These results are further extended to show that a combination of purine synthesis inhibition (MMF) along with RT is synergistic and significantly better than either treatment alone, in vivo. This establishes the clinical treatment potential of this combination and is further suggested by the reported high expression of TMPDH1 in aggressive brain tumor patients with lower survival. The experimental methodology is sound and the results most exciting and appropriate.

In the discussion section, the mechanism(s) by which GTP regulates RT-resistance is provocatively and perhaps correctly suggested to be a signaling process rather than simple nucleotide availability for DSD-repair. With regard to this suggestion, it would be most appropriate to look at the metabolism and the levels of AICA (ZMP), AMPK and effects with metformin in the presence of radiation, as well as MMF to determine if GTP signaling and feedback modulates these levels. With these minor studies in hand, the manuscript should be acceptable for publication, with significant contribution to the GBM literature and potential treatment modalities.

Reviewer #4 (Remarks to the Author); expert on one-carbon metabolism:

Attached

Comments on Zhou, W. *et al.* “Purine metabolism regulates DNA repair and therapy resistance in glioblastoma”

The paper describes a correlation of purine metabolites with radiation resistance in glioblastoma cells. The effect is demonstrated both in cell lines and xenograft models. The authors also show that the protection arises from a purine fueled increase in DNA repair. When de novo purine synthesis was inhibited, radiation sensitivity was restored. Overall, the findings are significant in understanding the differences in cellular response to radiation. The experiments are comprehensive and well executed; although possibly compromised by the possible loss of amplicons from these types of cells. However, there is one major oversight in the interpretation of the results. The authors cannot conclude the effect is due to de novo purine biosynthesis because their particular inhibitor affects both de novo as well as salvage synthesis. The paper should include a knockdown or pharmacological inhibition of a key enzyme necessary only for de novo biosynthesis. All they can claim is that the level of cellular purines affects radiation sensitivity. Likewise, the effect of radiation may compromise other cellular functions dependent on purines

Notably, Wang X., *et al.* *Nature Neuroscience* (2017) 20(5) 661-673 have reported a similar phenomenon with the brain tumor inducing cells (BTICs), where a clear effect on the de novo purine biosynthesis has been demonstrated. In my opinion the manuscript requires considerable revision before it would be considered suitable for publication in the Journal.

Specific comments.

1. (Results: Nucleotide metabolites correlate...) Can the authors please expand upon why no IDH1 mutations are good models of GBM? How might this impact the results?
2. (Results: Nucleotide metabolites correlate...) At the end of the third paragraph, the authors state that “Downregulation of the cytidine pathway was the third most-correlated metabolic pathway with RT-sensitivity...”; however, I do not see this represented in Figure 1C. Are the authors only showing those that are significant? Could *p*-values be added to Fig 1 C to demonstrate this?
3. (Fig. 1D.) It is somewhat perplexing that the total amount of AMP and GMP is constant between the radiation resistant and sensitive lines, while several other purine metabolites are different. Is there a metabolic or genetic reason for this observation?
4. (Results: Figure 2E) There seems to be a discrepancy between the 24 h time point result in Figure 2E (~70%) when compared to Figure 1B (~45%).
5. (Results: Supplementing nucleotide pools protects...) The last sentence in the first paragraph states “... reduced the DSBs presented 24 h after RT to near baseline levels”. It appears that 20% of cells still were scored as having positive γ -H2AX signal. This seems like a considerable amount of cells given that without RT, there were no (or minimal cells) having γ -H2AX signal. Please revise.
6. (Results: Inhibition...slows DSB repair...) The last sentence on page 9 states “...increased IMP levels by more than 10 fold and slightly increased ATP levels...” In the associated figure (Fig 3D), the increase in ATP abundance seems to be statistically significant (*p*-value associated with that read-out). I would suggest that a fold change value be included given the significance that has been assigned to the metabolite level (+MPA) relative to the control.
7. (Results: Figure 3D) Please extend the abundance axis marks, so we have a more clear idea of the ATP level in the MPA treated cells.
8. (Results: Inhibition...slows DSB repair...) Please define ER in the main text.
9. (Results: Inhibition...slows DSB repair...) Add errors to the ER values in the text since error was determined and presented in the associated figures. Also, the ratios used to compute ER seems to have been reversed between the Fig. 1B, C and D versus Fig. 2E and F. In Fig. 1B, C and D, was there a reason for following the survival of U118 for 6 hours but on DBTRG-05MG and GB1 for 8 hrs. The survival fraction curve for the control cells of U118 (Fig. 1B) has been fitted to a straight-line, has an exponential decay reaching a constant value been considered?

10. (Results: Figure 3F) Can the authors please state why there appears to be a substantial decrease in A172 resistance cells treated with nucleosides (lane 4)?
11. (Results: Inhibition...slows DSB repair...) What is the formulation of the +Nuc treatment exactly? Please provide details as to the composition in the main text and the formulation in the methods section. How do those levels compare to those in serum? Also, can you contribute the effect to decreased GMP/GTP when you supplement with guanosine? It might be converted to GMP directly. What would happen to the percent of positive γ -H2AX cells if you to remove guanosine from the nucleoside pool? Are you still able to rescue the cells from damage?
12. (Results: Figure 4A,B) In the test, the cells were allowed to "grow for 7-10 days before viability assessed"; however, the legend does not correctly state this. It states "...for Cell-Titer Glo assay 24 h post-RT (A&B)...". Additionally, "MSP12 and HF2303 neurospheres were treated as discussed above.." There is no discussion above. Please revise to be consistent with the text.
13. (Results: Inhibition...radiosensitizes...) The way the data are presented in the middle of the second paragraph, HF2303 data is presented first then MSP12. However, the figures are the other way around (MSP12 is Figs 4A and C; HF2303 is Figs 4B and D). Please switch around values for consistency in the presentation of data.
14. (Results: Inhibition...radiosensitizes...) The authors confused me on how they were able to go from cell growth to enrichment ratio. Please explain or convert numbers to be consistent with the way the data is presented in Figs 4A and 4B.
15. (Results: Purines, not...) In this section, a new cell line, GB-1, was introduced. Please explain why the cell line changed from MSP12 in the previous studies to GB-1.
16. (Results: Purines, not...) In the second sentence of the second paragraph it states "In the RT-sensitive GB-1 cell line, purines alone (adenosine and guanosine) promoted the repair of RT-induced DSBs nearly as much as pooled nucleosides." From my interpretation of Fig 5E, I see that only 60% rescue was obtained compared to 95-100% in the nucleotide pool. Even with error, this does seem to be more than "nearly as much". Please be more quantitative with the result and give a reason for why there appears to be a decrease in GB-1 cells.
17. (Results, Fig 7C) Please explain the discrepancy with tumor #3. This appears to be an outlier.
18. (Discussion/General Comment) In the second paragraph and mentioned throughout the text, the authors state that their data suggests "high rates of de novo purine synthesis". I can see where this could be misinterpreted. The de novo pathway (as illustrated in Fig 3A) starts with PRPP to make IMP; however, in your metabolomics studies, you cannot verify that this the case. It is possible that the salvage process is taking the free base and making the mono-phosphate form of the purine and from there, generating the triphosphate. I would highly recommend that the authors use a stable isotope label for glutamine and show how it is being incorporated into GMP/GTP and AMP/ATP. By doing this experiment, we are able to understand the original substrate for the purines and the process that RT-resistant or -sensitive GBM cells use to make purines.
19. The authors show the metabolic effect of MPA on GTP accumulation after 24 hours post treatment. Was the overall NADPH/NADP⁺ ratio and NADH/NAD⁺ assessed? Have the authors considered the effect on mitochondrial DNA replication and maintenance of the function owing to the requirement of GTP by the mitochondrial DNA polymerase.
20. For consistency, only one of the three phases should be used- intratumorally, Intra-tumoral, or *intra*-tumoral.
21. Fig. 3 B and D- data can be plotted to show y-axis end points.

Response to reviewers

1. To Reviewer 1:

“As an approach to overcoming the presumed intra-tumor heterogeneity in radiosensitivity existing in GBMs, this manuscript investigates purine synthesis as a determinant of GBM radioresistance. Towards this end, a panel of 23 established GBM cell lines are ranked according to their in vitro radiosensitivity as determined by clonogenic assay and subjected to metabolic analysis, which suggested that purines were related to GBM radioresistance. Inhibition of purine synthesis using mycophenolic acid (MPA) was shown to enhance radiosensitivity and replacement with exogenous nucleotides was shown to abrogate the radiosensitization. Similar results were obtained using 2 patient-derived GBM neurosphere cultures. In vivo experiments were then performed using flank xenografts grown from a GBM cell line and the neurospheres. Based on the data presented, the authors concluded that purine synthesis mediates the radioresistance of GBM and propose a clinical trial combining an MPA derivative and radiotherapy. However, there are a number of experimental deficiencies in this study (see below). Moreover, the relevance of the established cell lines to the biology/radiobiology of GBMs can be questioned. While the in vitro studies are aided by the addition of the 2 neurosphere cultures, all in vivo experiments are performed using flank xenografts, which do not account for the unique circumstances of brain microenvironment. The absence of an orthotopic model system questions the relevance of these results to GBMs and their therapy. ”

Reply: We thank the reviewer for the insightful comments and constructive suggestions. We agree that an orthotopic model is the optimal way to study GBM and had fortunately already begun to explore intracranial PDX models of GBM because they faithfully recapitulate much of the biology of patient GBM tumors. We chose GBM38, which is the most RT-resistant PDX model in the Mayo Clinic Brain Tumor PDX National Resource (PMID 31121035). Treatment with RT alone (median survival ~46 days) or MMF alone (median survival ~44 days) had only mild effects compared to control (median survival ~42 days). However, combined RT and MMF slowed tumor growth (as assessed by bioluminescent imaging, panels B and C, below) and significantly improved survival of tumor bearing mice (median survival ~62 days, panel D below). We have updated these data into new Figure 7A-D in the revised version of our manuscript.

Effect of MMF+RT in Intracranial Orthotopic PDX model (GBM38)

“Fig.1. The basis for the ranking of radiosensitivity among the established cell lines (1A) is unclear. It appears that for the majority of lines there is no statistically significant difference in Dmid. Whereas it seems reasonable to focus subsequent experiments on cell lines on the extremes of the measurement, the Dmid ranking does not appear to correspond to the actual survival curves shown for “sensitive” lines in figure 5. That is, U118MG, DBTRG-05MG and GB-1 are classified as sensitive according to Dmid, yet 6Gy results in a surviving fraction of 0.2 in U118MG (similar to U87 and A172, which are classified as resistant – figure 5), and almost 0.01 in the other 2 lines.”

Reply: We apologize for this lack of clarity and appreciate the opportunity to explain.

As we briefly noted in the text, Dmid is defined as the mean inactivating dose of radiation. Mathematically, Dmid is defined as the area under the clonogenic survival curve (Radiation research 99, 73-84 (1984). Similar to pharmacology studies, the area under the curve provides more information than a single data point such the IC₅₀ or the surviving fraction at a single radiation dose (<https://www.ncbi.nlm.nih.gov/pubmed/22066911>). We agree that the U118 MG cell line has a relatively high surviving fraction at 6 Gy compared to the other RT sensitive cell lines used. However, it has a much lower surviving fraction at 2 Gy compared to RT resistant lines (approximately 0.4-0.5, compared to 0.8-0.9 in U87/A172 lines), which is why the U118 MG curve appears to lack a “shoulder” at lower radiation doses. The Dmid incorporates all these pieces of information to give a composite readout of RT sensitivity, which is towards the lower end of the panel for the U118 line. Our basis for ranking cell lines was a simple rank-ordering

based on Dmid, and we did not perform statistical testing between cell lines. We have clarified the definition of Dmid, and that it represents the area under the curve, in the text (Line 102-103; Page 5).

“In figure 1D, it is stated that 2 resistant and 2 sensitive lines were subjected to metabolic evaluation after irradiation, yet only 2 columns were presented. Were the values averaged?”

Reply: The reviewer is correct. The data shown in the figure are indeed averaged.

The left column of Resistant is shown by the average fold changes of the two RT-resistant cell lines U87 MG and A172, whereas the right column of Sensitive is shown by the average fold changes of the two RT-sensitive cell lines, U118 MG and KS-1. We have clarified this point in the figure legend of Figure 1D.

“Fig.2. The alkaline comet assay is claimed to measure radiation-induced DSBs. This is inaccurate in that this assay primarily measures radiation-induced single strand breaks, which are a nonlethal event. To measure DSBs, it is necessary to use the neutral comet assay. This is significant in that the comet results were used to claim that the nucleosides do not affect the initial level of radiation-induced DSBs, but only influenced their repair. The rationale for performing the alkaline comet assay was that gammaH2AX analysis cannot distinguish between the induction and repair of DSBs, which is also inaccurate. gammaH2AX foci correspond to the initial level of DSBs at 0.5h after irradiation; their dispersal correlates with repair. In 2E-G, nucleosides were shown to decrease the initial level of gammaH2AX foci detected at 0.5h as well as those detected at times out to 24h. Thus, the data presented actually suggest that the nucleoside addition reduces the number of radiation-induced DSBs, which is in conflict with the conclusions derived from the comet assay. ”

Reply: We apologize for our imprecise language. We agree that the assay best suited to physically measure only DSBs is the neutral comet assay, while the alkaline comet assay reports on both DSBs and other non-lethal events such as single-strand breaks (<https://www.ncbi.nlm.nih.gov/pubmed/?term=26250399>). We have clarified our language regarding the alkaline comet assay in the manuscript and that we are using it as a composite readout of different types of physical DNA damage. In regards to the question of gamma H2AX foci at 30 minutes after RT, we agree with the reviewer that the abundance of gamma H2AX foci typically correlates with the number of DSBs at these earlier times in most cases. However, when early events in the DNA repair process are altered, this correlation is less certain (for example, when ATM is inactivated there are lots of DSBs but little gamma H2AX foci, <https://www.ncbi.nlm.nih.gov/pubmed/11571274>). Because numerous repair processes are activated within seconds to minutes after the induction of DSBs (<https://www.ncbi.nlm.nih.gov/pubmed/1677379>) and could potentially be regulated by nucleotide abundance, we chose to completely eliminate any DNA repair by performing the comet assays on ice. We have changed our language to more clearly state that nucleosides did not alter the induction of DNA damage (as opposed to DSBs) as measured by the alkaline comet assay. (Line 172-185; Page 8-9).

“Fig.4. Whether the differences between Control and MPA are statistically significant should be shown.”

Reply: The plots shown in previous figure 4 A&B (current figure 3G&H) are representative of 3 separate experiments (each experiment performed in technical triplicate) for HF2303 and two for MSP12. Thus, the error bars for individual points on the graph are from technical replicates of an individual experiment. Rather than showing statistical significance of technical replicates in the figure, which may be misleading, we have instead listed the average enhancement ratio and standard error of all experiments in the main text. We have added these enhancement ratios to the figure as well to provide additional clarity.

“Fig. 7F. For this analysis GBM is combined with low grade gliomas (LGG). To this point the entire paper had focused on GBM. The biology as well as the treatment of LGG and GBM are different. Moreover, if it was just GBM, the data would have to take into account percent resection, MGMT status and patient age, which are not accounted for in LGG analyses. The combination of these tumor types certainly complicates data interpretation and should be clearly justified.”

Reply: We appreciate these very good points.

The central issue here is that the “low grade glioma” and “GBM” TCGA datasets are molecularly heterogeneous with many different tumor types defined in each data set. With the hindsight provided by the molecular definitions in the 2016 WHO classification of diffuse gliomas and subsequent cIMPACT-NOW reports, we are better able to define the primary IDH wild type glioblastomas whose biology is the main focus of this manuscript. We aimed to exclude all IDH mutant tumors but to include IDH wt lower grade gliomas whose biology is similar to “traditional” glioblastoma.

Of the n=235 IDH-wt TCGA cases we initially used for survival analyses, n=137 were from the ‘Glioblastoma Multiforme’ study (and thus were grade IV by classic histopathologic criteria). A total of n=98 IDH wild type cases were from the ‘Lower Grade Glioma’ TCGA study (of note, all “classic” lower grade IDH mutant astrocytomas and oligodendrogliomas were excluded). Of these lower grade cases, 71 had either or both of *TERT* promoter mutation, the +7/-10 signature (reflecting gain of chromosome 7 on which *EGFR* resides and loss of chromosome 10 on which *PTEN* resides), and *EGFR* amplification. If one of these molecular events is present, it now satisfies the criteria for Glioblastoma, WHO grade IV (Brat DJ, Aldape K, Colman H, et al., *Acta Neuropathol.* 2018 Nov;136(5):805-810). Thus, while these tumors were classified as lower grade glioma by the TCGA, they are molecularly identical to GBM, have a dismal prognosis, and if diagnosed in 2020 would be classified and treated as GBM. We therefore kept these patients in our analysis.

This left n=27 cases of IDH wild type tumors from the lower grade glioma dataset that were either not profiled or were wildtype for *TERT* promoter mutation and/or did not harbor +7/-10 signature or *EGFR* amplification. Looking at these 27 cases more closely, the vast majority of them fall into the “Pilocytic astrocytoma-like” methylation group. This is a subgroup of IDH-wildtype diffuse gliomas that is clinically and molecularly distinct from “glioblastoma.” We therefore excluded these 27 patients from our analysis.

We were left with 208 patients with “molecular” IDH wild type GBM. Reanalysis of this purified cohort revealed similar findings as our initial analysis and is now included in Fig. 7E of the manuscript. We thank the reviewer for bringing up this point and for the opportunity to re-examine the patient cohort we analyzed.

The **previous** figures are shown as following:

The **new** updated figures are shown as following:

The mRNA expression of the rate-limiting enzymes of nucleotide pathways in 208 patients from the Pan-Cancer Atlas with newly diagnosed IDHwtGBM and the survival analysis.

To account for age and MGMT promoter methylation status as potential confounders in the survival analysis, we analyzed both of these in both the *IMPDH1* high and low groups. *IMPDH1* high tumors (median age 60, 39% MGMT promoter methylated) had nearly identical characteristics as *IMPDH1* low tumors (Median age was 59, 40% MGMT promoter methylated). Extent of resection was not readily available and this is a limitation of this analysis. We have noted the similarities in age and MGMT promoter methylation status between the *IMPDH1* groups in the text, as well as the limitation that some clinically important variables are missing.

2. To Reviewer 2:

“This is an interesting study, extending the previous work of Jeremy Rich and colleagues regarding the importance of purine metabolism in glioblastoma stem-like cells (Wang Nat Neurosci 2017), and targeting of pyrimidine synthesis to overcome resistance to drugs (Wang Sci Trans Med 2019).

Here, the authors report that inhibition of purine metabolism can lower radioresistance in glioblastoma cell lines, spheroids, and subcutaneously growing xenograft models in mice - by interfering with DNA repair. This is in principle a valuable extension of the previous work, and of high translational significance. The drug they use (MMF) is approved, widely used in the clinic, and even capable of passing the blood-brain barrier (a fact the authors should discuss!) -

which makes it to a very interesting candidate for future combination trials with radiotherapy in the clinic. The data appears solid, the manuscript is very well written, the statistics appears adequate and sound, and the conclusions well founded.”

Reply: We thank this reviewer for the positive comments.

“I have two major issues:

1.) It is well known today that radioresistance in vitro, and very likely also in non-orthotopic tumor models in vivo is a very poor readout for the "real" radioresistance seen in glioblastoma growing where it belongs (brain). There are several important cellular mechanisms of radioresistance that appear to be brain-specific (e.g., see Osswald et al., Nature 2015). Therefore, the authors MUST add data of patient-derived glioma stem-like cell lines (NOT U87, which is a very, very bad cell line for glioma research today), growing in the mouse brain - and confirm their radiosensitization findings with MMF. As discussed above, they are in a very good position: MMF is brain-penetrant, which is a rare feature for oncological drugs, so these are highly meaningful experiments that, if positive, would dramatically increase the value of this manuscript.”

Reply: Excellent point and we agree with the reviewer. Please see response to comment one to reviewer 1.

“2.) The supportive patient data (Fig. 7F) is interesting. I would like to see the same data for IDH-mutant gliomas. Is this mechanism of (potential) relevance here, too? - I am aware that IDHmut glioma models in vitro and in vivo are tricky, so I am not asking to provide experimental data - but in silico analyses from the TCGA and other databases are straightforward, and would also benefit this manuscript.”

Reply: We agree. Although our current work focuses on IDHwt GBM, whether these results hold true for the different biology of IDHmutant GBM is an interesting question. We identified 22 patients with IDHmutant GBM in the PANCAN dataset, which were either grade 4 by classic histology or grade 4 by virtue of homozygous deletion of CDKN2A/B (<https://www.ncbi.nlm.nih.gov/pubmed/31832685>). IMPDH1 expression was not prognostic in this group of patients, but it is difficult to draw concrete conclusions about this result given the small number of patients. Interestingly, high expression of DHODH, a rate-limiting step in *de novo* pyrimidine synthesis, was adversely prognostic in IDHmutant GBM. These data suggest that further investigation of pyrimidine biosynthesis could be interesting in gliomas with an IDH mutation. High expression shows an inferior survival in IDHmutGBM (Fig. 5 attached as following). We have incorporated these data (shown below) into Fig. S7A in the revised version.

A

The mRNA expression of the rate-limiting enzymes of nucleotide pathways in 22 patients from the Pan-Cancer Atlas with newly diagnosed grade 4 IDHmutGBM and the survival analysis

3. To Reviewer 3:

“This manuscript systematically demonstrates that radiation (RT) resistance in GBM cells results from an increased ability to repair RT-induced DSBs. Further, increased expression of IDH1 is associated with RT-resistance. The study also demonstrates that GBMs with depleted purines resulted in RT-sensitivity and that protection with added nucleotides in RT-sensitive GBMs facilitates repair of DSB and manifests resistance. In contrast, inhibition of de novo purine synthesis, particularly GTP synthesis via MMF, results in RT-sensitization of GBMs in cells as well as in patient-derived GBM neurospheres. These results are further extended to show that a combination of purine synthesis inhibition (MMF) along with RT is synergistic and significantly better than either treatment alone, in vivo. This establishes the clinical treatment potential of this combination and is further suggested by the reported high expression of TMPDH1 in aggressive brain tumor patients with lower survival. The experimental methodology is sound and the results most exciting and appropriate.”

Reply: Thank you for the encouraging comments.

“In the discussion section, the mechanism(s) by which GTP regulates RT-resistance is provocatively and perhaps correctly suggested to be a signaling process rather than simple nucleotide availability for DSD-repair. With regard to this suggestion, it would be most appropriate to look at the metabolism and the levels of AICA (ZMP), AMPK and effects with metformin in the presence of radiation, as well as MMF to determine if GTP signaling and feedback modulates these levels. With these minor studies in hand, the manuscript should be acceptable for publication, with significant contribution to the GBM literature and potential treatment modalities.”

Reply: This is a very interesting hypothesis that deserves further study. AICAR is detectable on our mass spectrometry method, so we reanalyzed our data to determine if it was affected by either RT or MPA. AICAR levels doubled following 24 h of MPA treatment, likely because AICAR is upstream of IMPDH (the target of MPA) in the *de novo* purine synthesis pathway. However, AICAR levels were not affected by RT either in the presence or absence of MPA. It is possible that this increased AICAR could be a mediator of the effects of MPA (possibly through regulation of AMPK, as the reviewer suggests). Our next manuscript aims to explain the mechanism by which GTP directly regulates DNA repair, and the AICAR/AMPK/mTOR signaling axis, and whether it is modulated by GTP will be fully explored. We do not intend to include these AICAR data in this manuscript, as we believe they are beyond the scope of the current work.

Effects of RT and MPA on AICAR. RT-resistant U87 cells were treated with MPA (10 µM) or vehicle control (Control) for 24 h and then either treated with 8 Gy radiation (RT) or mock irradiation (No RT). Two hours later, cells were frozen, metabolites were extracted and analyzed by mass spectrometry. Raw counts were normalized to No RT/Control, which was set to 1. Error bars indicate standard deviation of four biologic replicates

4. To Reviewer 4:

“The paper describes a correlation of purine metabolites with radiation resistance in glioblastoma cells. The effect is demonstrated both in cell lines and xenograft models. The authors also show that the protection arises from a purine fueled increase in DNA repair. When *de novo* purine synthesis was inhibited, radiation sensitivity was restored. Overall, the findings are significant in understanding the differences in cellular response to radiation. The experiments are comprehensive and well executed; although possibly compromised by the possible loss of amplicons from these types of cells. However, there is one major oversight in the interpretation of the results. The authors cannot conclude the effect is due to *de novo* purine biosynthesis because their particular inhibitor affects both *de novo* as well as salvage synthesis. The paper should include a knockdown or pharmacological inhibition of a key enzyme necessary only for *de novo* biosynthesis. All they can claim is that the level of cellular purines affects radiation sensitivity. Likewise, the effect of radiation may compromise other cellular functions dependent on purines.

Notably, Wang X., et. al. Nature Neuroscience (2017) 20(5) 661-673 have reported a similar phenomenon with the brain tumor inducing cells (BTICs), where a clear effect on the de novo purine biosynthesis has been demonstrated. In my opinion the manuscript requires considerable revision before it would be considered suitable for publication in the Journal.”

Reply: We thank this reviewer for these constructive suggestions to improve our manuscript. We agree that MPA/IMPDH inhibition could be exerting its effects by blocking purine salvage if there was substantial flux from hypoxanthine to IMP through HGPRT. We have performed several experiments to explore this possibility, the results of which suggest that *de novo* purine synthesis plays a greater role in mediating radiation resistance than does purine salvage. These include (1) tracing studies with ¹⁵N-amide glutamine showing that a substantial portion of the guanylate pool is derived from *de novo* synthesis, (2) showing that inhibition of GTP salvage (either through knockdown of *HPRT1* or media depletion of hypoxanthine) does not affect GBM RT sensitivity and (3) that inhibition of GARFT, an enzyme involved only in *de novo* purine synthesis, radiosensitizes GBM. These data are presented in greater detail in our point-by-point response below.

Specific comments:

“1. (Results: Nucleotide metabolites correlate...) Can the authors please expand upon why no IDH1 mutations are good models of GBM? How might this impact the results?”

Reply: IDHwtGBM (the focus of our study) is a separate molecular entity from IDH mutant GBM. These diseases have different natural histories, molecular features, metabolic phenotypes and responses to therapy. Because IDHwtGBM have a higher incidence (~90% of all GBMs), shorter survival time and characteristic radiation resistance, we focused on this entity. Determination of metabolic strategies to radiosensitize IDH mutant GBM would be of great interest. Our exploratory analysis (see response to reviewer 2) suggests that DHODH could be one such target. Another potential target is glutaminase, as mutant IDH tumors appear to rely on this enzyme to generate glutamate and glutathione because IDH-generated 2HG inhibits BCAT1/2 and limits glutamate production from transamination reactions (<https://www.ncbi.nlm.nih.gov/pubmed/30220459>).

“2. (Results: Nucleotide metabolites correlate...) At the end of the third paragraph, the authors state that “Downregulation of the cytidine pathway was the third most-correlated metabolic pathway with RTsensitivity...”; however, I do not see this represented in Figure 1C. Are the authors only showing those that are significant? Could p-values be added to Fig 1 C to demonstrate this?”

Reply: The reviewer is right. We only showed the metabolic pathways meeting significance criteria. We have indicated this in the figure legend. In the main text, we note that the cytidine pathway has a p value of 0.08 (Line 139-140; Page 7)

“3. (Fig. 1D.) It is somewhat perplexing that the total amount of AMP and GMP is constant between the radiation resistant and sensitive lines, while several other purine metabolites are different. Is there a metabolic or genetic reason for this observation?”

Reply: We do not have a clear explanation for this finding. Perhaps it suggests that nucleotides with higher energy charge are needed to mediate the RT response while the monophosphate forms are less important? This could either be due to the stored energy itself or the ability of triphosphate nucleotides to be converted into cyclic nucleotide signaling molecules or to directly stimulate signaling pathways.

“4. (Results: Figure 2E) There seems to be a discrepancy between the 24 h time point result in Figure 2E (~70%) when compared to Figure 1B (~45%).”

Reply: The gamma H2AX data shown in Fig.1B were collected by flow cytometry, whereas those shown in Fig. 2E were detected by immunofluorescence. The flow cytometry method has been reported to be less sensitive than counting gamma-H2AX foci by immunofluorescence microscopy (<https://www.researchgate.net/publication/221924513>). We have clarified the techniques used in the figure legends.

“5. (Results: Supplementing nucleotide pools protects...) The last sentence in the first paragraph states “... reduced the DSBs presented 24 h after RT to near baseline levels”. It appears that 20% of cells still were scored as having positive γ -H2AX signal. This seems like a considerable amount of cells given that without RT, there were no (or minimal cells) having γ -H2AX signal. Please revise.”

Reply: We apologize for the imprecise language, which we have now edited. The new sentence is shown as following: “Indeed, in all three sensitive cell lines, RT alone caused a peak of γ -H2AX foci within 30 min that did not return to baseline by 24 h, whereas treatment with exogenous nucleosides decreased γ -H2AX foci at 0.5, 2, 6 and 24 h following RT (Figs. 2E-G; Figs. S2A-C).”(Line 168-171; Page 8).

“6. (Results: Inhibition...slows DSB repair...) The last sentence on page 9 states “...increased IMP levels by more than 10 fold and slightly increased ATP levels...” In the associated figure (Fig 3D), the increase in ATP abundance seems to be statistically significant (p-value associated with that read-out). I would suggest that a fold change value be included given the significance that has been assigned to the metabolite level (+MPA) relative to the control.”

Reply: We added the fold change value of ATP in the text. The new sentence reads as following: “Treatment with a clinically-relevant concentration of MPA ($10 \mu\text{M}$)³⁶ reduced GTP levels by more than 10-fold, increased inosine monophosphate levels by more than 10-fold and increased ATP levels 1.2-fold, consistent with inhibition of IMPDH and little GTP generation from guanine (Figs. 3A-D).” (Line 207-210; Page 10).

“7. (Results: Figure 3D) Please extend the abundance axis marks, so we have a more clear idea of the ATP level in the MPA treated cells.”

Reply: Thank you for bringing this to our attention, we have made this correction in the revised manuscript.

“8. (Results: Inhibition...slows DSB repair...) Please define ER in the main text.”

Reply: In addition to our definition in the methods section, we have now also defined ER in the main text. Per reviewer’s suggestion, we defined ER in the main text as “ER of RT is defined as

Dmid control divided by Dmid treatment. ERs below 1 indicate radioprotection and above 1 indicate radiosensitization.” (Line 166-167; Page 8).

“9. (Results: Inhibition...slows DSB repair...) Add errors to the ER values in the text since error was determined and presented in the associated figures. Also, the ratios used to compute ER seems to have been reversed between the Fig. 1B, C and D versus Fig. 2E and F. In Fig. 1B, C and D, was there a reason for following the survival of U118 for 6 hours but on DBTRG-05MG and GB1 for 8 hrs. The survival fraction curve for the control cells of U118 (Fig. 1B) has been fitted to a straight-line, has an exponential decay reaching a constant value been considered?”

Reply: We have now added the errors to each ER value in the text (Line 211-222; Page 10-11). “Also, the ratios used to compute ER seems to have been reversed between the Fig. 1B, C and D versus Fig. 2E and F.” We think that the reviewer may be talking about Fig. 2B, C and D. For the ER value, it means radioprotecting if the value is below 1 (like the ER value of Nuc in Fig. 2B, C and D), whereas it means radiosensitizing if the value is above 1 (like the ER values of MPA1uM and 10uM compared to Control in Fig. 3E and F). We have clarified this point in the legend of Figure 2/main text (Line 166-167; Page 8).

Regarding the question of timing and curve fitting for the U118 MG cell line, we can offer several clarifying points. The X axis for figures 1B/C/D reflects dose of radiation (0, 2, 4, 6, 8 Gy), rather than time in hours. Survival for all cell lines was measured 10-14 days after radiation, which is standard for clonogenic survival assays. The reviewer is correct that we only included doses of radiation 0-6 Gy for the U118 MG line, while DBTRG-05MG and GB1 went from 0-8 Gy. This is because the dose of 8 Gy killed all the cells in the U118 MG line making it impossible to calculate a “surviving fraction.” Using a surviving fraction of 0 is incompatible with the linear-quadratic curve fitting equation that we used for all cell lines. The reason the linear-quadratic curve fit for U118 MG appears to be dominated by the linear component may be due to its lower surviving fraction at 2 Gy (discussed above in response to reviewer 1) and the absence of data at 8 Gy.

“10. (Results: Figure 3F) Can the authors please state why there appears to be a substantial decrease in A172 resistance cells treated with nucleosides (lane 4)?”

Reply: We apologize that we confused the reviewer. “a substantial decrease in A172 resistance cells treated with nucleosides” was caused by our setting of Y axis. We previously set 0.9 as the minimum number of the Y axis but didn’t label that in the figure, which made the bar in Lane 4 (the exact number is ~0.9) look very low. We have modified and labeled clearly in both Fig. 3 E&F in the revised manuscript

“11. (Results: Inhibition...slows DSB repair...) What is the formulation of the +Nuc treatment exactly? Please provide details as to the composition in the main text and the formulation in the methods section. How do those levels compare to those in serum? Also, can you contribute the effect to decreased GMP/GTP when you supplement with guanosine? It might be converted to GMP directly. What would happen to the percent of positive γ -H2AX cells if you to remove guanosine from the nucleoside pool? Are you still able to rescue the cells from damage?”

Reply: Excellent comments.

The concentration for each component in the 100x pool is as following: cytidine 0.73 g/L, guanosine 0.85g/L, uridine 0.73g/L, adenosine 0.8g/L and thymidine 0.24 g/L. The working

concentration we used in this paper is 8x, which leads to a final concentration of 80-240 μM for the various nucleoside species. We used these high concentrations as relatively blunt tool to perturb intracellular nucleotide pools. Whether nucleosides from the serum, where concentrations typically range from 1-10 μM , or tumor interstitial fluid, where nucleosides such as uridine can reach concentrations up to 100 μM (PMID 30990168), can promote RT resistance is a very interesting question that warrants further study. We have a complete description of the composition of the nucleoside pool to the main text (Line 160-162; Page 8 in main text and “Cell culture and reagents” part in Supplemental).

[REDACTED]

“12. (Results: Figure 4A,B) In the text, the cells were allowed to “grow for 7-10 days before viability assessed”; however, the legend does not correctly state this. It states “...for Cell-Titer Glo assay 24 h post-RT (A&B)...”. Additionally, “MSP12 and HF2303 neurospheres were treated as discussed above..” There is no discussion above. Please revise to be consistent with the text.”

Reply: We apologize for this oversight. The previous Fig. 4A&B are now Fig. 3 G&H in the revision. We reedited the figure legends as “**(G&H)** HF2303 or MSP12 neurospheres were treated as the timeline shown in Fig. S3E. In brief, cells were treated with nucleosides or MPA, and retreated with nucleosides 2 h before RT. Cells were replated to the 96-well plate (2000 cells/well) 24 h post-RT and cell viability were detected by the Celltiter-Glo kit approximately 7 days after replating. Note: Fig. G and H are representative figures from 2-3 repeated experiments.....”

“13. (Results: Inhibition...radiosensitizes...) The way the data are presented in the middle of the second paragraph, HF2303 data is presented first then MSP12. However, the figures are the other way around (MSP12 is Figs 4A and C; HF2303 is Figs 4B and D). Please switch around values for consistency in the presentation of data.”

Reply: Per reviewer’s suggestion, we now present HF2303 first then MSP12 in the newly updated Figures.

“14. (Results: Inhibition...radiosensitizes...) The authors confused me on how they were able to go from cell growth to enrichment ratio. Please explain or convert numbers to be consistent with the way the data is presented in Figs 4A and 4B.”

Reply: We apologize for the confusion. We have now explained figure legend that the enhancement ratio (ER) for sphere-forming assays is calculated as the GI50 of the control-treated cells divided by the GI50 of the treated cells.

“15. (Results: Purines, not...) In this section, a new cell line, GB-1, was introduced. Please explain why the cell line changed from MSP12 in the previous studies to GB-1.”

Reply: MSP12 and HF2303 are both patient-derived neurosphere culture models of GBM, while GB-1 is an adherent immortalized GBM line (which we profiled in Figure 1 and used as a model of a RT sensitive line in Figure 2). We wanted to discriminate between the roles of pyrimidines and purines in both types of cultured GBM models, so we chose GB-1 and HF2303. We have clarified this point in the text. In the time since our initial submission, we have also repeated

these experiments in a third cell line (the RT-sensitive adherent immortalized GBM cell line DBTRG-05MG) with consistent results (below). We have updated the Fig. 5 in the revised manuscript to reflect these new results.

Purines but not pyrimidine can rescue irradiation-induced DNA damage

“16. (Results: Purines, not...) In the second sentence of the second paragraph it states “In the RT-sensitive GB-1 cell line, purines alone (adenosine and guanosine) promoted the repair of RT-induced DSBs nearly as much as pooled nucleosides.” From my interpretation of Fig 5E, I see that only 60% rescue was obtained compared to 95-100% in the nucleotide pool. Even with error, this does seem to be more than “nearly as much”. Please be more quantitative with the result and give a reason for why there appears to be a decrease in GB-1 cells.”

Reply: We apologize for this imprecise language. We have clarified in the text that we have two models where purines provide a nearly full rescue and one where the rescue is around 64%. (Line 272-283; Page 13).

“17. (Results, Fig 7C) Please explain the discrepancy with tumor #3. This appears to be an outlier.”

Reply: We agree that this tumor appears to be an outlier. It is possible that this reflects the heterogeneity inherent in patient-derived models such as HF2303.

“18. (Discussion/General Comment) In the second paragraph and mentioned throughout the text, the authors state that their data suggests “high rates of de novo purine synthesis”. I can see where this could be misinterpreted. The de novo pathway (as illustrated in Fig 3A) starts with PRPP to make IMP; however, in your metabolomics studies, you cannot verify that this the case. It is possible that the salvage process is taking the free base and making the mono-phosphate form of the purine and from there, generating the triphosphate. I would highly recommend that the authors use a stable isotope label for glutamine and show how it is being incorporated into GMP/GTP and AMP/ATP. By doing this experiment, we are able to understand the original substrate for the purines and the process that RT-resistant or -sensitive GBM cells use to make purines.”

Reply: We appreciate the reviewer highlighting this important issue and giving us the opportunity to clarify. We agree that the primary tool we use to inhibit *de novo* GTP synthesis (MPA and MMF) can also inhibit the generation of GTP from salvaged hypoxanthine through

the HGPRT reaction and thus can inhibit both *de novo* and salvage GTP synthesis. We have attempted to address this oversight in several ways.

First, we performed stable isotope tracing using ^{15}N -amide glutamine in the RT-resistant U87 GBM cell line. Within 4 h, nearly half of the GMP, UMP and CMP pools were generated through *de novo* synthesis (AMP was lower, possibly due to the larger adenylate pool size). We have included these data as supplemental figure S3. While we would have liked to pursue additional tracer studies with longer incubations of tracer in order to reach steady state, our lab and our metabolomics core has been shuttered due to COVID-19 and such studies are not possible for the foreseeable future. Even with these limitations, these experiments demonstrate significant activity of *de novo* nucleotide synthesis in this model.

***De novo* nucleotide synthesis is active in RT-resistant GBM cells**

We have also asked whether inhibition of hypoxanthine salvage affected the phenotype of RT resistance in a similar fashion to MPA treatment. Depletion of hypoxanthine (from 30 μM in control media to absent in experimental media, both with dialyzed FBS) had no effect on the radiosensitivity of U87 model. We included these data as Figure S3F in the revised manuscript.

F
Hypoxanthine deprivation does not impact RT sensitivity in U87 MG cells

Using a separate patient-derived neurosphere model we silenced *HPRT1*, the gene encoding HGPRT, which salvages both hypoxanthine (to form IMP) and guanine (to form GMP). Knockdown of *HPRT1* had no effect on the radiosensitivity of the MSP12 model. We incorporated these data into Fig. S3G&H in the revised manuscript.

G**H**
Knockout of HPRT1 does not radiosensitize GBM neurospheres

[REDACTED]

“19. The authors show the metabolic effect of MPA on GTP accumulation after 24 hours post treatment. Was the overall NADPH/NADP⁺ ratio and NADH/NAD⁺ assessed? Have the authors considered the effect on mitochondrial DNA replication and maintenance of the function owing to the requirement of GTP by the mitochondrial DNA polymerase.”

Reply: Unfortunately, our mass spectrometry method did not detect NADPH/NADP⁺ nor NADH. The hypothesis regarding whether modulating GTP levels affects DNA repair by regulating the mitochondrial DNA polymerase is an interesting one, and we hope to fully explore this and other mechanistic questions in future work.

“20. For consistency, only one of the three phases should be used- intratumorally, Intra-tumoral, or intratumoral.”

Reply: Thank you. We carefully double checked and made all the phrases or words in the manuscript consistent.

“21. Fig. 3 B and D- data can be plotted to show y-axis end points.”

Reply: Per reviewer’s suggestion, we fixed this formatting issue.

REVIEWERS' COMMENTS:

Reviewer #1 (Remarks to the Author):

The authors have adequately addressed my comments and criticisms

Reviewer #2 (Remarks to the Author):

The authors have responded well to my comments and suggestions. Particularly the new data confirming the results with orthotopic PDX models has greatly increased the value of this study.

Reviewer #3 (Remarks to the Author):

The revised manuscript is acceptable for publication. We look forward to the next paper in the series.

Reviewer #4 (Remarks to the Author):

I am satisfied with the revised ms that all my comments have been addressed.

Response to Referees

Reviewer #1 (Remarks to the Author):

The authors have adequately addressed my comments and criticisms

Reviewer #2 (Remarks to the Author):

The authors have responded well to my comments and suggestions. Particularly the new data confirming the results with orthotopic PDX models has greatly increased the value of this study.

Reviewer #3 (Remarks to the Author):

The revised manuscript is acceptable for publication. We look forward to the next paper in the series.

Reviewer #4 (Remarks to the Author):

I am satisfied with the revised ms that all my comments have been addressed.

Author Reply

We sincerely thank the reviewers for the time and thoughtful suggestions to improve this manuscript.